# Culling corallivores improves short-term coral recovery under bleaching scenarios

Jacob G. D. Rogers ⬤ [1,2] ✉ & Éva E. Plagányi ⬤ [2]

Management of coral predators, corallivores, is recommended to improve coral cover on tropical coral reefs under projected increasing levels of accumulated thermal stress, but whether corallivore management can improve coral cover, which is necessary for large-scale operationalisation, remains equivocal. Here, using a multispecies ecosystem model, we investigate intensive management of an invertebrate corallivore, the Crown-of-Thorns Starfish (*Acanthaster cf. solaris*), and show that culling could improve coral cover at sub-reef spatial scales, but efficacy varied substantially within and among reefs. Simulated thermal stress events attenuated management-derived coral cover improvements and was dependent on the level of accumulated thermal stress, the thermal sensitivity of coral communities and the rate of corallivore recruitment at fine spatial scales. Corallivore management was most effective when accumulated thermal stress was low, coral communities were less sensitive to heat stress and in areas of high corallivore recruitment success. Our analysis informs how to manage a pest species to promote coral cover under future thermal stress events.

---

[1] School of Mathematics and Physics, University of Queensland, Brisbane, QLD 4072, Australia. [2] CSIRO Oceans and Atmosphere, Brisbane, QLD 4072, Australia. ✉email: Jacob.Rogers@csiro.au

Failure to redress contemporary upper ocean temperature rise will result in annual bleaching events and the onset of long-term reef degradation by 2050 across most of the world's coral reefs[1–3]. Managing portfolios of refugial regions for corals, such as Australia's Great Barrier Reef[4], are likely to be pivotal in providing coral reefs opportunities for either natural- or artificially mediated acclimation and/or adaptive capacity in the face of unprecedented rates of warming[4–7]. In these systems, local-scale interventions may reduce large-scale pressures that are less amenable to direct management, thereby providing more opportunity for coral reefs to adapt and acclimate[5,8–10].

Invertebrate coral predators, corallivores, are targeted by coral reef management authorities globally[11–14]. Removal of an invertebrate corallivore (*Coralliophila abbreviata*) has been demonstrated to support coral thermal resistance and recovery following bleaching events[14]. The removal of another corallivore (*Acanthaster spp.*) is similarly posited as a viable means to increase coral reef resilience in other systems[8,9]. In general, corallivore management is suggested to be a fundamental consideration in restoration initiatives (e.g. refs. [15–17]). Unfortunately, the operationalisation of corallivore management remains challenging, largely due to the extensive effort and resources required to scale up management over larger areas[18–20] and limited quantifiable success in doing so[20,21]. Despite this, recent strategic efforts to control the corallivorous Crown-of-Thorns Starfish (CoTS, *Acanthaster c.f. solaris)*, on the Great Barrier Reef (Fig. 1) have been correlated with increased coral cover throughout a period encompassing two major bleaching mortality events and elevated CoTS numbers[12]. CoTS exhibit highly selective behaviour for coral species[22,23], mobility (maximum escape response velocity indicative of up to 250–520 m/day depending on substrate and prey availability though likely <19 m/day displacements)[24,25] and indifference to prey colony size[26]. Therefore, they pose a serious ecological risk to reef resilience to bleaching events by impeding recovery potential[26,27]. Current approaches to managing coral cover trajectories directly are limited[9,12,28], but corallivore management under thermal stress shows promise[12,14,29,30]. However, we must understand how management intervention interacts with corallivores, corals and thermal stress to shape intervention outcomes and effectiveness. *Acanthaster* species undergo large population fluctuations (irruptions but more commonly termed outbreaks) and feeding by individuals during outbreaks has caused widespread loss of coral cover across many Indo-Pacific coral reefs[30–32]. Their destructive potential has made them of critical interest to management authorities[31], most prominently in Australia and Japan (e.g. refs. [11,12,18,19,21,32]). Here we developed a fine-scale, mechanistic model of reefs representing the key drivers of CoTS and their interaction with both their coral prey and management interventions. This enables us to quantify the potential of local-scale corallivore management to mitigate the thermal bleaching impacts that are globally projected to become more frequent.

Multispecies ecosystem models, such as Models of Intermediate Complexity for Ecosystem Assessment (MICE), are well-suited to informing effective management strategies in the face of natural and anthropogenic changes[33–35]. MICE are increasingly applied across a multitude of systems to provide information for tactical management advice[36–39]. MICE are discrete-time models involving systems of difference equations primarily fitted through maximum likelihood estimation (e.g. refs. [37,39]) and solved numerically. A restricted focus of MICE to just the key components of the ecosystem, and their interactions, allows these models to handle uncertainties whilst accounting for an ecosystem context[33,35]. This makes them ideal for devising and evaluating tactical management interventions in a similar fashion to fisheries management (e.g. refs. [36,38]).

In the current study, we developed and fitted a MICE model at the sub-reef scale on the central Great Barrier Reef to data spanning two bleaching events and including CoTS management efforts[12,17,28,37]. We captured CoTS dynamics across four size-based age classes as well as two characteristic groups of coral prey. Our model not only represented CoTS removals, but had an underlying (validated) population dynamics representation that dynamically updated estimated changes in population abundance due to the net effect of growth, recruitment, mortality and culling. In addition to interactions amongst CoTS, coral and management, we also resolved the concomitant impacts of thermally induced bleaching and cyclones upon corals and CoTS management. We show that CoTS management could improve coral outcomes, but efficacy was dependent on local factors.

## Results

Using available data from 2013–2018 within the Cairns section of the Great Barrier Reef (Fig. 2), we validated the MICE model (fit statistics summarised in Table 1). We then used our model to test four alternative thermal stress and two management intervention scenarios under three different levels of coral adaptive and acclimation capacities—a total of 24 different ecological-management realisations (Table 2). Scenario projections were averaged over 80 simulations which produced substantially different outcomes. Results are summarised in terms of management efficacy, which we define here as the consequence of management intervention (CoTS culling) relative to no intervention. Management sites 2 and 7 were included here to demonstrate characteristic outcomes of bleaching events over 2015–2017 within our dataset and for their near median response to thermal stress perturbation scenarios (Figs. 3, 4). For transparency, all model fits and outputs for each management site under each scenario are provided (as per Figs. 3, 4) in the Supplementary Information (Supplementary Figs. 1–37). Details of each model fit (one for each adaptive and acclimation capacity) are summarised in Table 1 and delineated by the management site in Supplementary Table 1. Management-induced differences in coral cover and CPUE over model projections are also provided (Supplementary Figs. 38–43). Accumulated thermal stress maps[40] were used to validate model fitted values for management sites and corresponded well to these (Supplementary Figs. 44, 45, Supplementary text 1). Model parameters, variables and where applicable, their sources, are provided in the Supplementary Information (Supplementary Tables 2–6). Within each model, 68 parameters were fitted (Supplementary Table 6).

**Management effectiveness varied**. The impact of culling CoTS was heterogenous across sub-reef management sites in terms of coral cover and catch-per-unit-effort rates (e.g. Figs. 3, 4). All management sites were sensitive to accumulated thermal stress whereby increased accumulated stress and decreased adaptive and acclimation capacity resulted in increased levels of coral mortality (Supplementary Figs. 1–37). Culling reduced catch-per-unit-effort rates and thereby indirectly improved coral cover at some sites but not all (Supplementary Figs. 40–42). The benefit of intensive corallivore control relative to no control increased sigmoidally, with trajectories demonstrating asymptotic behaviour over time (Fig. 5). Thus, the median improvement in coral cover across the management sites we considered saturated at ~2% over our 10-year projections (Fig. 5), while the mean saturated at 4.2% (Supplementary Figs. 38–40). The increased mean compared with median derived coral cover benefit was due to positive skewness in the impact of management intervention, which ranged from an improvement of ~0.2% through to as much as ~23% total coral cover (Supplementary Figs. 38–40). This suggests the mean was

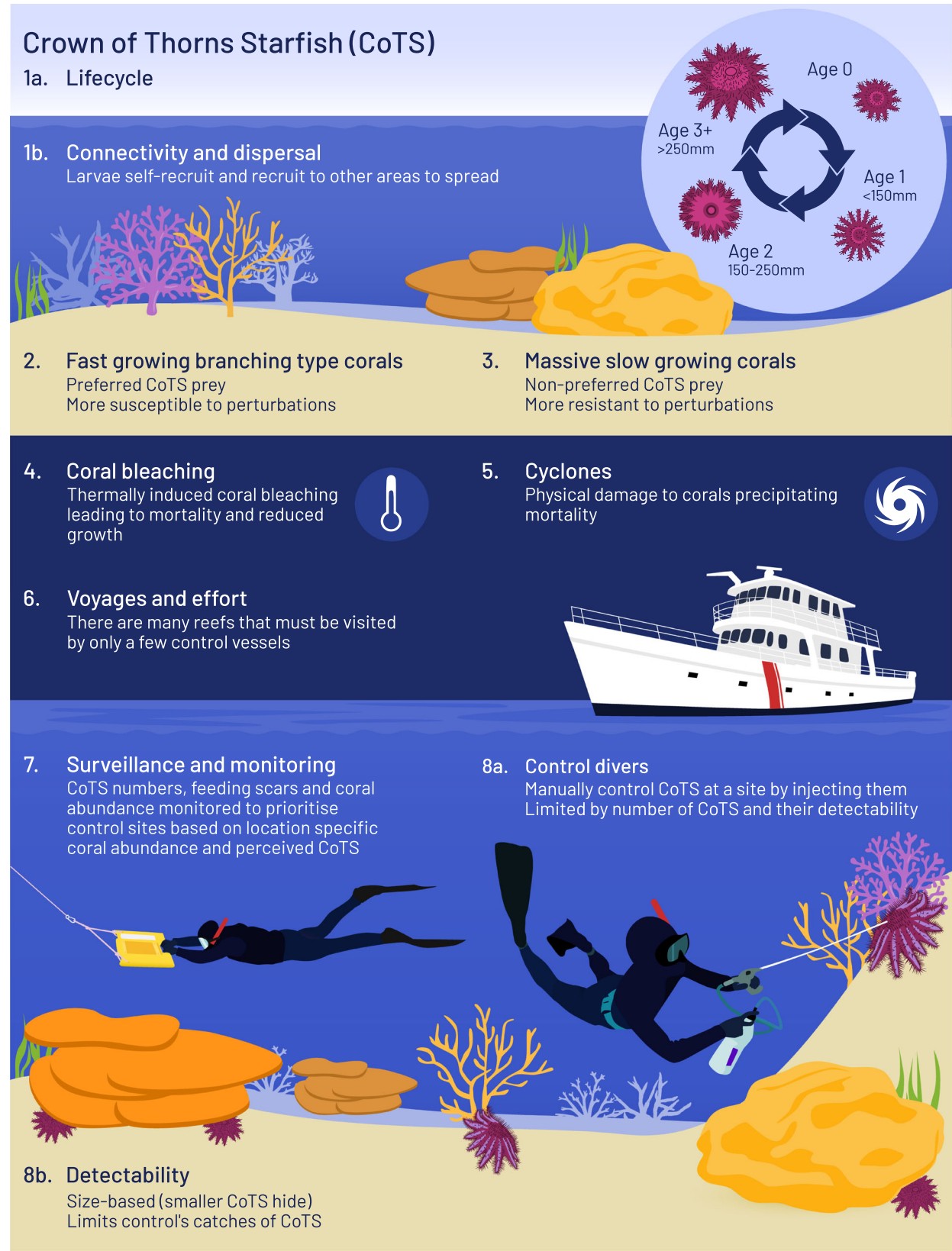

not as informative as the median, but it also indicates there is a greater opportunity to increase efficacy than there is for reduced efficacy. Substantial positive skewness and saturation of management efficacy were also found in management-induced reductions of catch-per-unit-effort rates (Supplementary Figs. 41–43). Here management-induced differences ranged from a catch-per-unit-

effort rate reduction of 0.01 through to 0.25 CoTS.min$^{-1}$. This meant that management was suggested to saturate at a median reduction of 0.05 or a mean reduction of ~0.08 CoTS.min$^{-1}$. Simulation of thermal stress events resulted in attrition of management-derived benefit of coral cover (Fig. 5) and CoTS catch-rate reductions (Fig. 6) at sites. However, attrition was

**Fig. 1 Overview of crown-of-thorns (CoTS) biology, threats and management on the Great Barrier Reef (GBR). 1a** Management of CoTS on the GBR is structured around the abundance of different size classes of CoTS. **1b** Reproductively mature age-2 and age-3+ CoTS spawn their gametes into the environment and rely upon prevailing conditions to achieve fertilisation and distribute larvae. **2–3** Pacific CoTS preferentially target faster-growing corals (e.g. *Acropora*, *Pocillopora* and *Montipora*) over coral taxa characterised by slower growth rates and massive morphologies (e.g. *Porites*) which are generally consumed less than expected based on their abundance. **4–5** Coral are subject to environmental perturbations with preferred taxa generally more susceptible to cyclone events and thermally induced bleaching. **6** On the GBR, CoTS outbreaks may span large geographical regions encompassing many reefs across which a limited number of control vessels and divers must be distributed. **7–8a** Current management of the species primarily entails the deployment of divers at prioritised locations where they manually cull individual CoTS to reduce coral consumption by the species. **8b** Principal known drivers of heterogenous manual removal impacts are CoTS size and age as well as population density. Detectability of smaller individuals is a key constraint of contemporary management intervention. Cumulatively, corallivore intervention success at the regional scale first demands success at the finer scales at which they operate. Strategising intervention requires resolving analogous processes **1–5** and **7–8** to achieve broader scale results through **6**.

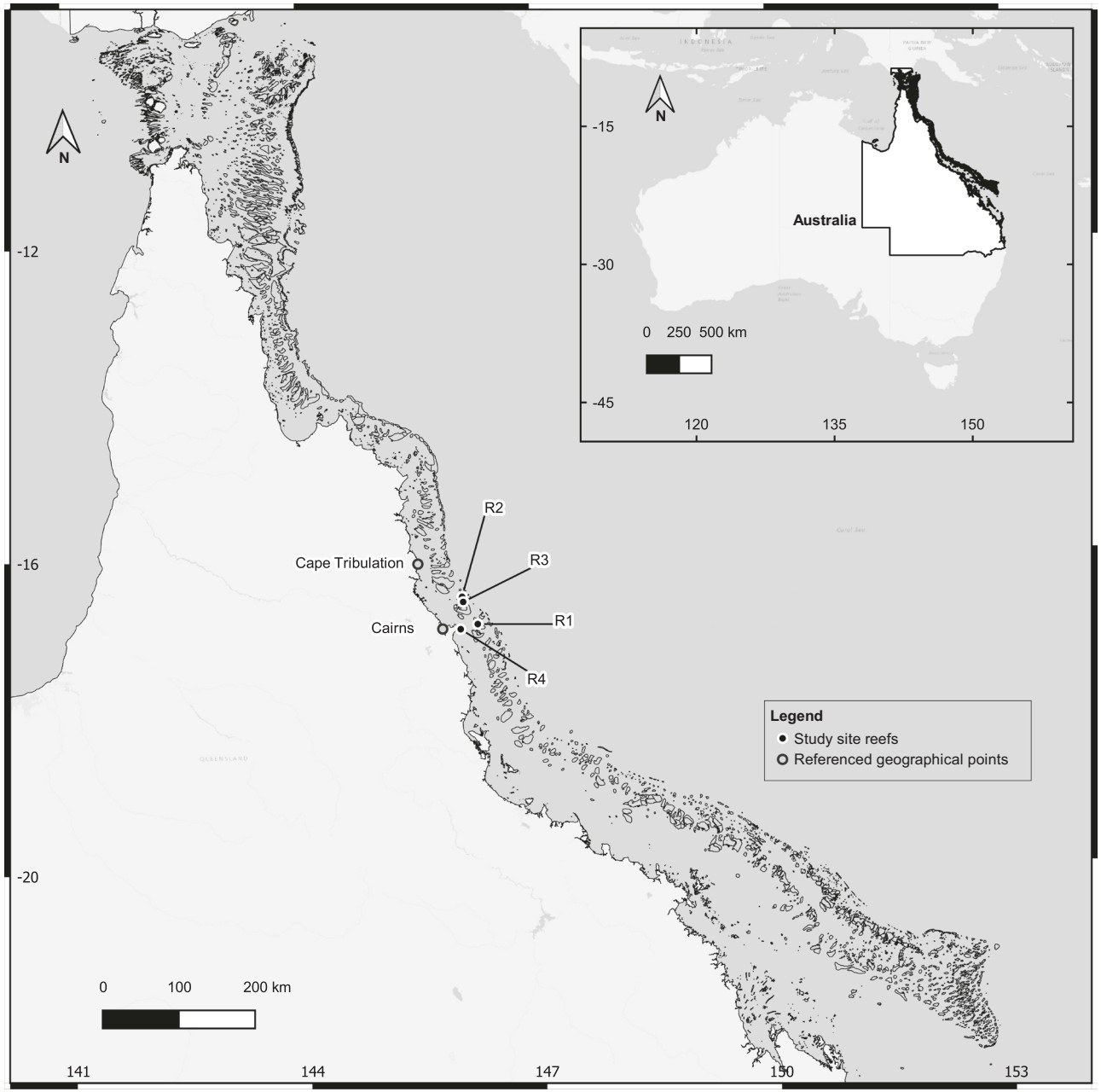

**Fig. 2 Study sites on Australia's Great Barrier Reef (GBR).** Map delineating reefs on which management control sites used in this study were located and key geographical points referenced in this study. Prefix "R" denotes reef, for example, R1 is reef 1. Developed in QGIS with GBR features dataset[113].

**Table 1 Summary of likelihood contributions arising from each data source (coral cover data and catch-per-unit-effort data (CPUE)) and encompassed penalty terms.**

|  | Model likelihood | | |
| --- | --- | --- | --- |
|  | *A* = 0 | *A* = 2.5 | *A* = 5 |
| **-lnL(Total)** | −40.8655 | −40.8803 | −40.8865 |
| -lnL(Overall Coral Cover) | −196.111 | −196.131 | −196.14 |
| -lnL(Overall CPUE) | 138.832 | 138.835 | 138.837 |
| **Penalty terms** | | | |
| Stock-recruitment penalty | 16.3327 | 16.3344 | 16.3356 |
| Catch scaling penalty | 0.0808 | 0.0813 | 0.0809 |

Parameters were fitted simultaneously with each model adaptive and acclimation capacity scenario (*A* = {0, 2.5, 5}) containing 68 estimated parameters. Given each model contains the same number of parameters the likelihood is used to discern relative model parsimony. Overall, *A* = 5 was the most likely model though this was negligibly so and all models were similar in their ability to describe the data. A full breakdown of likelihood contributions and computed data variability arising from each management site is available in Supplementary Table 1. Parameter estimates and their standard deviations are provided in Supplementary Table 6. Headings within table are presented in bold.

**Table 2 Summary of modelled scenarios.**

| Property | Values |
| --- | --- |
| Management control | No visits |
|  | Monthly visits (start of each month) |
| Thermal stress perturbation | DHW = 0 (None) |
|  | DHW = 4 (Mild) |
|  | DHW = 7 (Moderate) |
|  | DHW = 10 (High) |
| Adaptive/acclimation capacity | *A* = 0 (No adapt.) |
|  | *A* = 2.5 (Low adapt.) |
|  | *A* = 5 (Medium adapt.) |

Accumulated thermal stress is represented in terms of degree heating weeks (DHW). There were a total of 24 unique scenarios modelled. This are such that for each management control option, each thermal stress level and each adaptive/acclimation capacity is modelled (2 × 4 × 3 = 24 total scenarios). Headings within table are presented in bold.

dependent on the level of thermal stress, adaptive and acclimation capacity and CoTS recruitment connectivity (Fig. 5 and Supplementary Table 6). Under moderate to high levels of thermal stress (DHW = 7 and DHW = 10), intensive corallivore management was only beneficial to corals with the greatest adaptive and acclimation capacity (*A* = 2.5 and *A* = 5; Fig. 5). Sites with greater CoTS recruitment (Supplementary Table 6) derived increased benefit from intensive corallivore management; for example, comparing Management Site 6 (where the model suggested was high recruitment; Supplementary Fig. 5) with Management Site 13 (low recruitment; Supplementary Fig. 11). Cumulatively, management efficacy depended on local factors at the sub-reef scale, as derived benefits varied across sites within the same reef. However, the persistence of coral cover benefits is indicative that intensive corallivore management improves short-term coral recovery under the thermal stress, management intervention and adaptive and acclimation scenarios considered here (Figs. 5, 6).

**Management most effective under mild thermal stress.** Management intervention was most effective at locations where thermal stress was lower or if corals were better able to buffer against thermal stress. Corallivore management under higher levels of thermal stress (DHW = 7 and DHW = 10) was constrained by the low coral cover, which reduced the derived benefit

of management intervention. This was evident in coral cover trajectories (e.g. Figs. 3, 4) due to bleaching-induced mortality. Reduced rates of catch-per-unit-effort reduction were due to prey limitation. This is because prey limitation led to an increase in CoTS mortality rates. The former manifested a threshold drop (Fig. 5), while the later decayed less abruptly (Fig. 6). Under the no thermal stress scenario (DHW = 0), coral cover was predictably higher than in the alternative scenarios where a bleaching event was simulated (e.g. Figs. 3, 4). However, the coral cover benefit derived from intensive corallivore management under the mild thermal stress scenario (DHW = 4) was initially similar and then surpassed that of the no bleaching event scenario (Fig. 5). This was despite negligible difference between the mean and median catch-per-unit efforts rates between DHW = 0 and DHW = 4 scenarios (Fig. 6). This is because modelled CoTS populations were demographically similar and of similar abundance under both scenarios. Consequently, the greater benefit was most likely due to a reduction in coral growth rates due to inter- and intraspecific competition amongst corals. This is supported by management site trajectories having lower coral abundance as accumulated thermal stress increased (Supplementary Figs. 1–11). This dynamic was consistent across different thermal adaptive and acclimation capacity scenarios (Supplementary Figs. 12–37). However, increased capacity precipitated a sooner emergence of mild thermal stress as the most effective application of intensive corallivore management (Fig. 5)—expediting coral recovery.

**Sensitivity for model outputs over 2013–2018.** Contrasting fitted model trajectories that included CoTS control, with simulations that did not (i.e. simulated Management Sites as if control had not taken place), suggest that there was a median improvement of ~1% in coral cover (mean was higher at ~2.1%) over the five years of 2013–2018 (Fig. 7a). The bleaching event of 2016 and 2017 substantially reduced the signal of CoTS control (in terms of coral cover) in modelled trajectories (Fig. 7a, b). Prior to the bleaching events, manual control is suggested by the model to have improved median coral cover over years 2013–2015 by ~1.9% (mean was ~3.3%) but this was quickly reduced to a median of ~0.7% (mean was ~1.9%). Predominantly due to the bleaching events, we found that the no control scenario indicated that median coral cover for the group of Management Sites in 2018 (relative to 2013) would likely have declined by ~23% without CoTS control but by a lesser amount of a median of ~18% with control (Fig. 7b, c). In terms of relative change at individual Management Sites over 2013–2018, the median of the differences was ~3% (Fig. 7d). That is, considering the bleaching events in 2016 and 2017, CoTS control is suggested by the MICE to have increased coral cover by 3% relative to no-control over the timeframe.

**Discussion**
Numerous studies recommend corallivore control to improve coral cover under increasingly uncertain coral reef futures (e.g. refs. [12,14,30]). Although there has been mixed success in controlling CoTS throughout the Indo-Pacific[12,20,21,41], we found management improved modelled coral cover across all management sites. Moreover, our study suggests that the level of management-induced coral cover improvement increases rapidly at the start, but is bounded and conditional on local sub-reef-scale factors. In terms of coral cover, contrasting success[12,41] and failure[21] may be explained by the spatial scale and level of local ecological integration into management. To achieve the best ecological outcomes, corallivore control must be tactically distributed based on where it will be most effective[20]. Our study identified the drivers of heterogeneity—and management efficacy

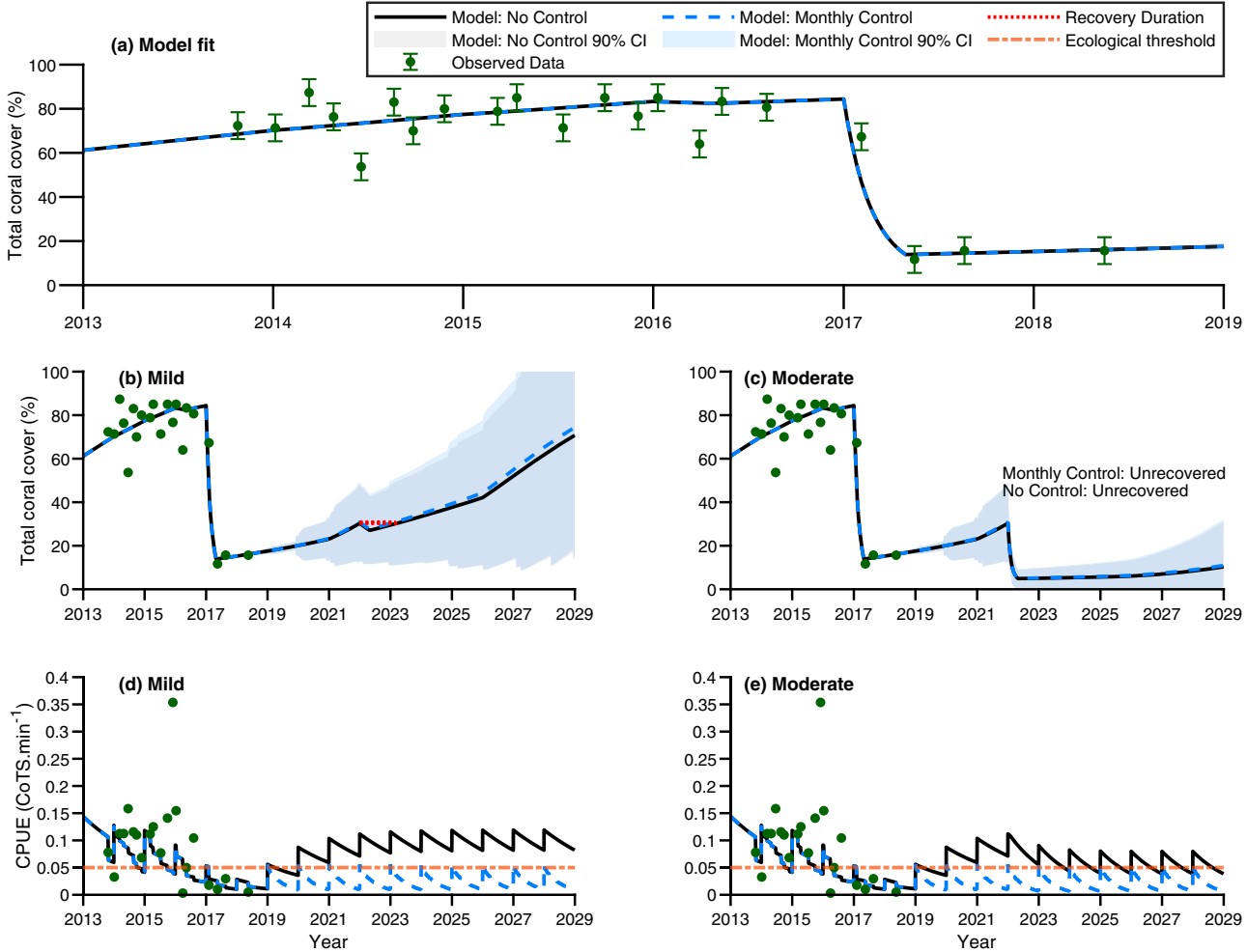

**Fig. 3 Management site 2 model fits and total coral cover (%) and catch-per-unit-effort (CPUE; CoTS.min⁻¹) of crown-of-thorns starfish (CoTS) trajectories under no manual control vs monthly manual control given no adaptive capacity ($A = 0$).** In all panels, black solid lines define the no control scenario and the blue dashed line defines the controlled scenario. **a** The model fit to total coral cover for the management site (data points plotted with error bars depicting ± 90% CI; green points and error bars). Each data point constitutes a single observation. Variance for the site was calculated at the Maximum Likelihood Estimate (MLE) which was simultaneously fitted to $n = 539$ observations (both coral cover and CoTS CPUE, see Supplementary Table 7). The 90% CIs were calculated based on the variance obtained for Management site 2's coral cover series and the number of observations in the series $n = 21$. **b**, **c** Total coral cover trajectories under different thermal stress levels expressed as Degree Heating Weeks (DHW $\in \{4, 7\}$) simulated in year 2022. Mean trajectories are presented ± 90% CI depicted as errors bands ($n = 80$ simulations; grey shading is a 90% CI error bands for the no control scenario, blue shading is 90% CI for the controlled scenario). If coral cover recovery to pre-perturbation levels is not observed by year 2029 the management scenario is denoted, 'Unrecovered'. A lack of perturbation-induced mortality is denoted by, 'Recovery not applicable'. **d**, **e** CPUE trajectories under DHW = 4 and DHW = 7 events. CPUE is a management-based measure of CoTS abundance conditional on demographics and detectability. Error bars are not displayed in **b**–**e** to simplify display. Variability in each management scenario's trajectory due to stochastic tropical cyclones impacts over years 2018–2029 is indicated by shaded uncertainty bands and was limited in CPUE trajectories. The ecological threshold of 0.05 CPUE above which CoTS consumption exceeds coral growth based on a coral cover of 35% is plotted[63] (orange dash-dot line). CPUE sawtooth curve patterns are due to model population dynamics as individuals annually become detectable to the management program. Source data (model outputs) provided.

—under thermal stress to be: (1) the adaptive and acclimation capacity of assemblages, (2) their thermal history and (3) the sub-reef distribution of successful corallivore recruits.

Coral assemblages can broadly adapt and acclimate in response to successive thermal stress events[42,43]. This can vary greatly at finer taxonomic scales due to a range of interacting intrinsic and extrinsic factors[44,45]. At high levels of thermal stress, we found control was of limited efficacy in improving coral cover where the adaptive and acclimation capacity of the assemblage was low. Conversely, efficacy improved with increased adaptive and acclimation capacity of corals and/or decreased thermal stress. This suggests that, depending on prevailing within-site conditions, management success could range from negligible (e.g. ~0.5% improvement in coral

cover) to highly successful (~23% improvement in coral cover). Extrinsic factors, such as solar radiation, local hydrodynamics, habitat features and water quality influences the thermal responses of corals[42,46,47]. The interplay of these with intrinsic factors—for example, the composition of coral assemblages, holobionts present, energy reserves and thermal stress histories—leads to fine-scale variability in assemblage thermal response/s both within and among reefs[44–46]. Based on our modelling results, the way these factors interact to confer thermal resilience to local assemblages parametrises potential management success under different thermal scenarios. That is, site-specific thermal susceptibilities contributed to substantial differences in the efficacy of local corallivore control. Greater coral taxa diversity than we considered here is likely to

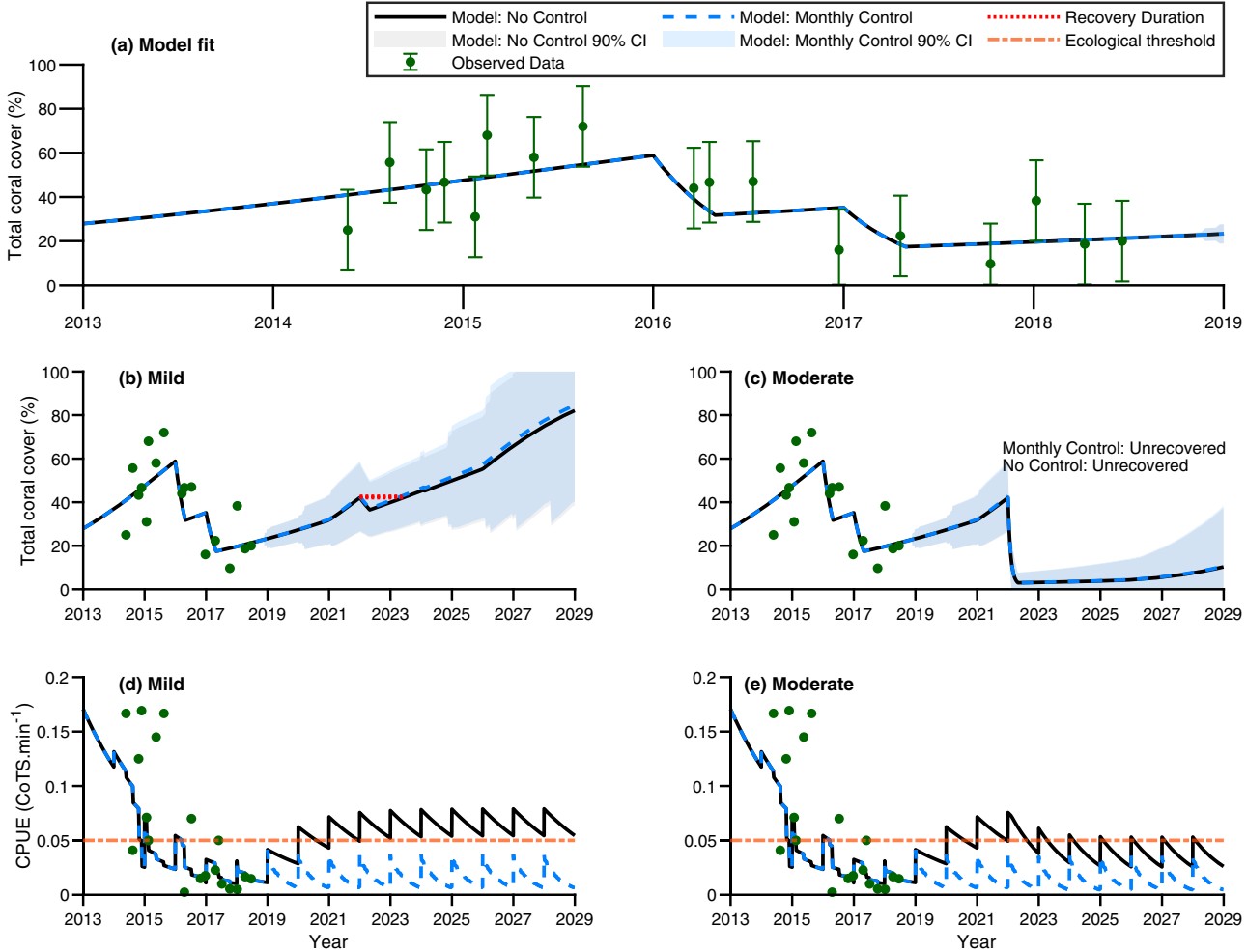

**Fig. 4 Management site 7 model fits and total coral cover (%) and catch-per-unit-effort (CPUE; CoTS.min$^{-1}$) of crown-of-thorns starfish (CoTS) trajectories under no manual control vs monthly manual control given no adaptive capacity ($A = 0$).** In all panels, black solid lines define the no control scenario and the blue dashed line defines the controlled scenario. **a** The model fit to total coral cover for the management site (data points plotted with error bars depicting ±90% CI; green points and error bars). Each data point constitutes a single observation. Variance for the site was calculated at the Maximum Likelihood Estimate (MLE) which was simultaneously fitted to n = 539 observations (both coral cover and CoTS CPUE, see Supplementary Table 7). The 90% CIs were calculated based on the variance obtained for Management site 7's coral cover series and the number of observations in the series n = 17. **b, c** Total coral cover trajectories under different thermal stress levels expressed as Degree Heating Weeks (DHW ∈ {4, 7}) simulated in year 2022. Mean trajectories are presented ±90% CI depicted as errors bands (n = 80 simulations; grey shading is a 90% CI error bands for the no control scenario, blue shading is 90% CI for the controlled scenario). If coral cover recovery to pre-perturbation levels is not observed by year 2029 the management scenario is denoted, 'Unrecovered'. A lack of perturbation-induced mortality is denoted by, 'Recovery not applicable'. **d, e** CPUE trajectories under DHW = 4 and DHW = 7 events. CPUE is a management-based measure of CoTS abundance conditional on demographics and detectability. Error bars are not displayed in **b–e** to simplify display. Variability in each management scenario's trajectory due to stochastic tropical cyclones impacts over years 2018–2029 is indicated by shaded uncertainty bands and was limited in CPUE trajectories. The ecological threshold of 0.05 CPUE above which CoTS consumption exceeds coral growth based on a coral cover of 35% is plotted[63] (orange dash-dot line). CPUE sawtooth curve patterns are due to model population dynamics as individuals annually become detectable to the management program. Source data (model outputs) provided.

drive more disparate bleaching responses—and management efficacies—among sites[44,48].

Regionally, corallivore management based on recruitment connectivity between reefs has demonstrated promise[12,19,49,50]. Our results suggest that efficacy may be improved through prioritising management sites based on recruitment connectivity and settlement success, particularly when this overlaps with higher levels of adaptive and acclimation capacity. Our model identifies sub-reef larval sinks as a key variable in management efficacy, but this must be interpreted contextually with model caveats. This is because successful larval recruitment regulates and influences emergent corallivorous adult populations within management sites in our model. Whilst this is consistent with

observed limited realised CoTS mobility (sensu displacement) and homing behavioural patterns[24,25], the importance of sub-reef larval sinks could plausibly be influenced by adult movements among management sites[24,25,27,51] and/or variation in the development of juveniles[52,53]. Adult movement can lead to localised outbreaks[27,51] and is a catalyst in broader outbreak dynamics[50]. Variation of juvenile CoTS growth rates may delay recruitment to catchable (and detectable) size classes and influence density estimates[52,53]. However, high juvenile mortality (a likely bottleneck[54]), a lack of concordant in situ observations[55–57] and the coupling of larvae detection rates with those of later age-1 individuals[58] obfuscate and suggest prolonged juveniles phases are difficult to detect and/or may have limited prevalence within

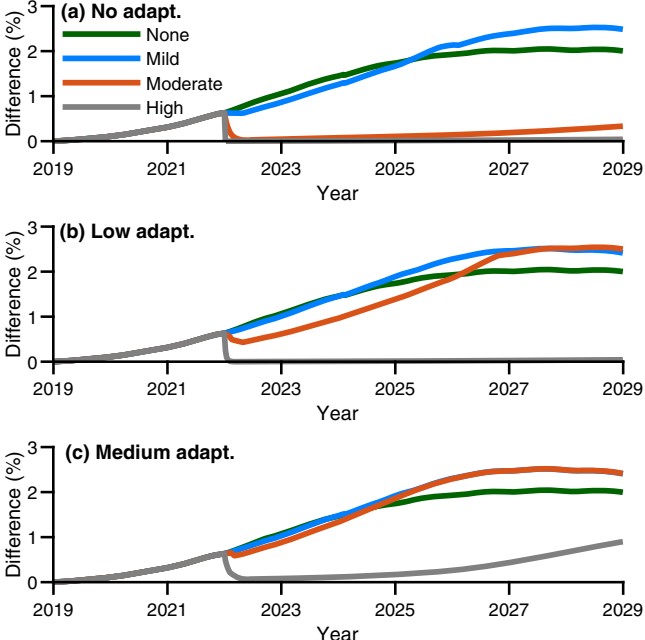

**Fig. 5 Adaptive capacity vs median improvement in coral cover due to manual control.** The median impact that monthly manual control makes across all sub-reef management sites expressed in terms of the difference in coral cover (%) between control and no control over a 10-year projected period. Different line colours correspond to different modelled levels of accumulated thermal stress experienced by corals. The green curve corresponded to no stress (Degree Heating Weeks (DHW) = 0), the blue curve corresponded to mild stress (DHW=4), the orange curve corresponded to moderate stress, (DHW=7) and the grey curve corresponded to high accumulated thermal stress (DHW=10). Figure panels vary in terms of modelled coral adaptive and acclimation capacity. **a** no adaptive and acclimation capacity ($A = 0$). **b** low adaptive and acclimation capacity ($A = 2.5$). **c** medium adaptive and acclimation capacity ($A = 5$). Increasing adaptive and acclimation capacity improved management-derived improvements in coral cover across higher levels of thermal stress. Mean was of similar dynamics, but attained higher values. Source data (model outputs) provided.

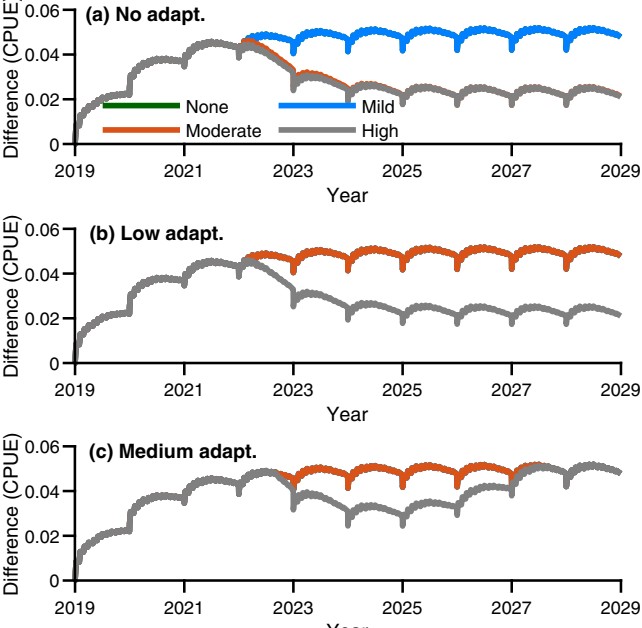

**Fig. 6 Adaptive capacity vs median reduction in crown-of-thorns starfish (CoTS) catch-per-unit-effort rates (CPUE, CoTS.min⁻¹).** The median impact monthly manual control makes across all sub-reef management sites expressed in terms of the difference in catch-per-unit-effort rates (CPUE) between control and no control over a 10-year projected period. Different line colours correspond to different modelled levels of accumulated thermal stress experienced by corals. The green curve corresponded to no stress (DHWs = 0), the blue curve corresponded to mild stress (DHW = 4), the orange curve corresponded to moderate stress, (DHWs = 7), and the grey curve corresponded to high accumulated thermal stress (DHWs = 10). Figure panels vary in terms of modelled coral adaptive and acclimation capacity. **a** no adaptive and acclimation capacity ($A = 0$). **b** Low adaptive and acclimation capacity ($A = 2.5$). **c** Medium adaptive and acclimation capacity ($A = 5$). Increasing levels of accumulated thermal stress reduced CoTS catch rates across different adaptive and acclimation capacities. Mean was of similar dynamics but attained higher difference values. Source data (model outputs) provided.

typical population dynamics. Aggregative dynamics of adults may have also precipitated increased catch rates via the development of aggregations for spawning and/or feeding—essential features of the species' biology[24,50,59]. Aggregative behaviours at the reef scale (i.e. across multiple sites)[27]—such as during outbreaks—and/or prolonged juvenile phases which would theoretically allow 'banks' of juveniles to build up, would likely reduce the role of sub-reef larval settlement patterns in the model. Nonetheless, our model results underscore the complex dynamics of sub-reef larval sinks[60] and adult movement behaviours[24,25]. Sub-reef larval sinks are found here to be a driver of management efficacy with implications for management and future work.

Management was most effective when targeting areas of high corallivore recruitment success under mild to no thermal stress and/or when focused on coral assemblages with higher levels of adaptive and acclimation capacity. Our results strongly support that the succession and responses of coral assemblages to disturbance regimes and histories be monitored and integrated into management site prioritisation schemes to facilitate increased efficacy under climate change[20,44,46,47]. Understanding site-specific thermal responses could drastically improve post-disturbance management by characterising expected efficacy[20]. For example, perturbations that result in variation of suitable CoTS settlement habitat may modify larval sinks[60] and how this

trades off with the coral cover. If deployed strategically and concentrated at sites where they are most beneficial, we demonstrate that direct control of CoTS could substantially improve coral cover relative to an uncontrolled scenario within 10 years—even under typical cyclone regimes (maximum value of 23% with 75th percentile of ~4% vs. median of ~2%). We support this by our findings that the management program operated across multiple sites partitioning effort among sites where they were of great impact (e.g. >10% improvement relative to no control) and where the model suggested there was a limited benefit (e.g. ~0.5% improvement relative to no control).

More generally, limited management efficacy within sites largely denuded of coral allows an intensification of corallivory on live corals in sparsely populated assemblages[27,29,61,62]. Under low or high coral cover, management efficacy was limited due to ecological constraints that reduced coral growth at high or low abundance[63]. This lowered the level of coral cover, whereby coral growth was outpaced by corallivore consumption[63]. Although not explicitly modelled here, such constraints may include competition, disease, lower recruitment/reproductive success, as well as a loss of fish-derived nutrients and/or herbivory[64]. Assemblages that maintained intermediate levels of coral cover (such as those with greater adaptive and/or acclimation capacity under thermal stress) facilitated increased efficacy as management lowered net

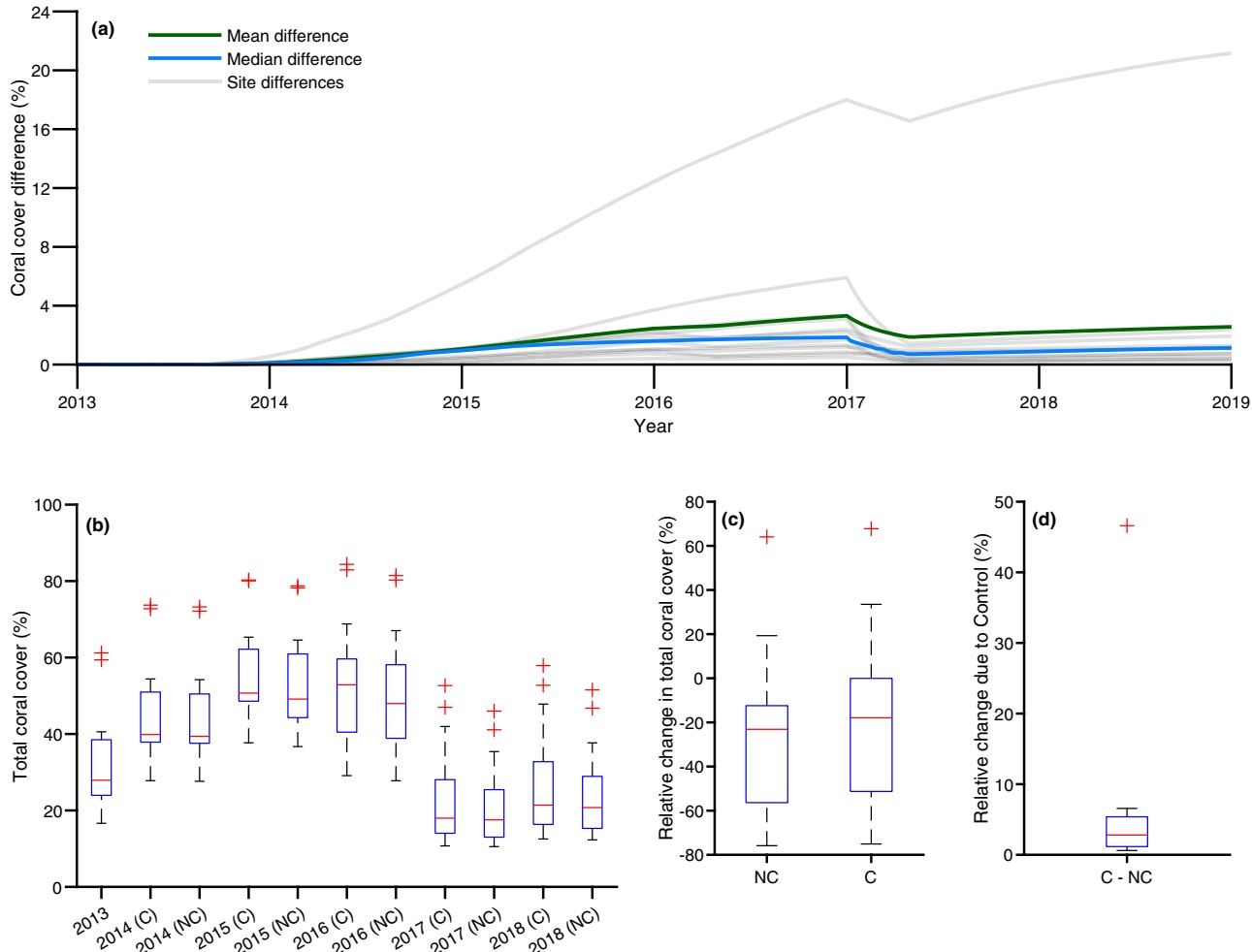

**Fig. 7 Sensitivity for model outputs over the data period of 2013–2018.** Figure is based on model outputs over 2013–2018 (June used as observed data ranged from mid-2013 to mid-2018 for most sites) which used undertaken control efforts by the Crown-of-Thorns Starfish (CoTS) Management Program as input (e.g. site visits and dive time) to inform potential coral trajectories in its absence. Sensitivity considered no adaptive and acclimation capacity ($A = 0$). Abbreviation 'C' denotes 'Control' (CoTS culling) and 'NC' denotes 'No Control'. Boxplots are based on model outcomes for the modelled management sites ($n = 13$). **a** Summary of differences in coral cover between management scenarios of no manual control and inputted observed manual control over years 2013–2018. Difference represents the (absolute) difference in coral cover expressed as a percentage as opposed to a proportional difference. Both mean (green line) and median (blue line) differences are plotted alongside the management-induced difference at each Management Site considered (grey lines). **b** Box and whisker plot indicating median coral cover model output (middle line), the 25th and 75th percentiles (the box) and any outliers (crosses) over years 2013–2018 with and without control. **c** Box and whisker plot indicating the model-suggested relative change in coral cover for 2018 relative to 2013 with and without control, positive values indicate an increase, negative values indicate a decrease and 0 indicates no change. NC: lower adjacent (lower whisker) −75.9%, 25th percentile (lower box) −56.4%, median (centre line) −23.1%, 75th percentile (upper box) −12.4%, upper adjacent (upper whisker) 19.3%. C: lower adjacent (lower whisker) −75.1%, 25th percentile (lower box) −51.3%, median (centre line) −17.9%, 75th percentile (upper box) −0.1%, upper adjacent (upper whisker) 33.5%. **d** Box and whisker plot indicating model-suggested relative change over 2013–2018 due to operation on the CoTS control program. Lower adjacent (lower whisker) 0.6%, 25th percentile (lower box) 1.2%, median (centre line) 2.8%, 75th percentile (upper box) 5.4%, upper adjacent (upper whisker) 6.6%. Source data (model outputs) provided, includes boxplot details.

consumption rates most relative to net coral growth rates. For coral restoration projects, intensification of corallivory means that coral outplanting may be more effective within sites that have reasonable coral cover as opposed to denuded locations[15,65], although such dynamics depend on corallivore populations and their local persistence (e.g. refs. [51,52]).

At the scale of reef networks, larval supply and connectivity drive the propagation of adult CoTS populations between reefs[28,49]. The present study identifies larval connectivity and settlement success at the sub-reef scale as essential variables in optimising the distribution of limited management resources and scaling manual control across larger spatial areas. There is, however, a paucity of understanding of how connectivity, larval

behavioural factors and habitat suitability operate to characterise successful settlement distributions at the sub-reef scale[54,60]. Resolving these relationships and their climate sensitivity will be essential in further refining the ecological basis of the current reef prioritisation protocol.

As a sensitivity we compared projected scenarios to the modelled trajectories over the data fitted period with and without CoTS control. We found the benefit of control persisted under the sensitivity although it was reduced over time (median improvement in the coral cover of ~1% over 5 years). The MICE suggested this to be due to the impacts of the 2016 and 2017 bleaching events. These events reduced the median of control-mediated improved coral cover by ~1.2%. Overall, the MICE

from 2013 to 2018 was consistent with the findings of our projection scenarios for 2019 to 2029 (improved up to a median of ~2%, high levels of thermal stress reduced detectable management signal).

Ideally, a before-after-control-impact (BACI; or similar) study (e.g. ref. [66]) would contribute to the understanding of CoTS control on coral trajectories. An empirical BACI analysis should be a priority for future CoTS GBR control work and could potentially be achieved by leveraging and combining multiple sources of data (e.g. ref. [67]) such as from the Australian Institute of Marine Science Long Term Monitoring Program data and recent Integrated Pest Management CoTS control program. Based on our modelling study, it is strongly recommended that care is taken in how to define 'control' sites and how well they are likely to track an impact site where CoTS management takes place. The present modelling suggests that the signal of CoTS control is heterogenous depending on local factors and sensitive to perturbation (bleaching). The 2016 and 2017 bleaching events have also been suggested by Westcott et al.[12] to have reduced the full benefit of CoTS control in terms of coral cover. A BACI analysis would provide additional insight into how CoTS control influences coral cover trajectories.

While our work was based on a contextualised dataset, we expect our results to have broader geographical relevance. It is, however, important to carefully consider generalisation. We note that most of our understanding comes from *A cf. solaris*[31] which is found through the Pacific and along the west Australian coast and that differences in outbreak severity and propensity may differ amongst *Acanthaster spp*[68]. This requires further research to understand if, how, and what species differences influence population dynamics and impacts[31]. We also mark that coral growth rates vary by coral species and by location which may influence modelled recovery rates[37,69]. The degree to which recovery rates are impacted will primarily depend on coral growth rates compared to CoTS consumption rates (sensu[63]). Given our model structure (e.g. CoTS preference for faster-growing corals, sequential depletion of coral taxa along preference gradients and bleaching susceptibility of coral prey) is consistent with that of systems in which *A. cf. solaris* are problematic[31,70], we expect robustness in the principle of CoTS management to reduce bleaching consequences; but, we also highlight the importance of carefully considering local conditions and the intrinsic capacity of corals to recover over a given timescale. Relative to intrinsic, local, coral recovery capacity, we expect our findings around CoTS control to be of general transposability within the range of *A. cf. solaris*. In relation to other species of invertebrate corallivore (e.g. gastropods), we highlight that their control is likely to also benefit from identifying local-scale principles[14,16], but may encounter greater difficulty in up-scaling culling efforts given the often relatively large scales of CoTS control efforts (e.g. refs. [12,21,71]).

Our results have implications for managers of tropical reef ecosystems globally when faced with managing the impacts of damaging corallivores and climate change. We found through ecological modelling, that improving coral cover under differing levels of accumulated thermal stress was positively related to coral assemblage adaptation and acclimation capacities and their thermal history, as well as the distribution of successful corallivore recruitment. These factors—and improvements in coral cover from better management– varied within and among reefs. We recommend incorporating fine-scale, sub-reef differences in climate-sensitive factors to effectively distribute limited management resources[4–6].

## Methods
Our model focused on the trophic interactions among CoTS and two groups of coral within a feedback loop with natural and anthropogenic forcing. Our model draws on accepted features of the published dynamics described by Morello et al.[37], Condie et al.[28] and Condie et al.[17], but is a substantial advance in terms of adding spatial structure and coupling with climate variables. Here we have resolved a fine spatiotemporal model structure, developed a novel recruitment formulation for

CoTS, integrated tactical management control dynamics and incorporated the impact of broad-scale drivers upon the population dynamics of corals and CoTS at the local scale. Our model is formally fitted to a subset of the CoTS control program data described by Westcott et al.[12]. We operationalised our model as a tactical and strategic tool to inform how CoTS management strategies interact with alternative disturbance and ecological realisations at the sub-reef scale, the scale at which management operates.

**Data**. We fitted our model to a subset of four reefs from the dataset described by Westcott et al.[12], which were consistently and intensively managed (for a map with reef locations see Fig. 2). We restricted our focus to a subset to avoid parametrisation of reef and management site dynamics. Thus, ~39% of site visits were concentrated over the 13 management sites we considered, with a mean of $20.73 \pm 5.5$ (mean ± standard deviation) visits across the time series relative to a mean visitation rate of $12.23 \pm 4.7$ (mean ± standard deviation) for the rest of the sites. Each reef in the subset contained two or more management sites where each site was visited at least 18 times. The subset was used because it contained sufficient data for estimating the 11 model parameters for each management site. Across included sites were a range of CoTS densities, coral abundances and disturbance histories[12,72,73]. Given the intensity with which these sites were managed, they therefore provided us with a valuable opportunity to formally fit the interactions between management intervention, coral abundance and CoTS dynamics in the presence of regional sequential bleaching events.

**Model spatial structure and ecological components**. Spatially, we considered a circular 300 km region of the Great Barrier Reef centred between Cairns and Cape Tribulation, and resolved at a daily timescale and a sub-reef spatial scale, matching the scale at which observed data were resolved[12,19]. Reefs were randomly generated as points to capture possible spatial correlation in disturbance impacts between nearby reefs, as well as to allow variability in reef locations. Coral, CoTS and disturbance dynamics within the management sites of each reef were resolved relative to a 1 ha focal region. That is, each management site was captured as a 1 ha area representative of the whole site. In the Pacific, *Acanthaster spp.* disproportionally target faster-growing corals, predominantly *Acropora*, *Pocillopora* and *Montipora*[22]. Coral taxa characterised by slow growth rates and massive morphologies, such as *Porites*, are generally consumed less than expected based on their abundance[22] and are thus non-preferred prey. The two modelled coral groups were the fast-growing favoured prey items of CoTS, and the slower-growing non-preferred prey. Processes resolved in the model included reproduction, density dependence, the effect of bleaching and cyclonic disturbances on corals and the impact of manual control (culling) upon CoTS and coral dynamics.

**CoTS population structure**. We used an age-structured approach to model CoTS population dynamics. We defined our age classes to encapsulate plausible size-at-age variation due to plastic growth. This was achieved through linking catch size classes of the management control program[19] to age classes through size-age relationships developed from observations spanning multiple environmental realisations, manipulated scenarios and methodologies[55,70,74,75]. Delayed growth in juvenile CoTS due to deferral of their switch to coral prey or composition of their pre-coral diet, may induce variability in the size-at-age of juveniles[52,53]. However, the population-level consequences of prolonged juvenile phases are not easily observed nor understood. For example, juveniles are subject to high mortality rates in situ, delayed growth may reduce lifetime fitness and there have been no observations of juveniles during spawning periods that would indicate protracted juvenile phases[55–57]. Consequently, suggests size-at-age is—due to an early life history mortality bottleneck or otherwise—predominantly concordant with growth curves of the literature[55,70,74,75] and the size classes we have used here. Age classes comprised annual 0, 1, 2 and 3+ groups, with 3+ being an absorbing class – once there, they stay there. Age-0 (<1 year of age) individuals were assumed to escape control program efforts due to their cryptic behaviour and small size[76–78], age-1 (1–2 years of age) corresponded to individuals <150 mm diameter, age-2 (2–3 years of age) individuals were 150–250 mm diameter and age-3 + (≥3 years of age) individuals were >250 mm diameter. We used empirically derived estimates of size-age detectability to model the number of CoTS within a population that would be available to control on a particular day. This accounted for the fact that some CoTS would still be undetected and remain on the reef. To this extent, we note that CoTS are voracious predators and it is unlikely that large numbers of undetected individuals would not alter coral trajectories[63,79]. Individuals in the three age classes were detected by management control activities at rates of 19%, 82% and 82% respectively[63,76].

Detection rates (as perceived on SCUBA) are predominantly a function of CoTS size and population abundance[63,76]. Recent studies have found little evidence for coral cover[76] or reef structural complexity[80] influencing detection rates (using SCUBA). This is likely due to divers being able to search the reef matrix for hidden CoTS unlike manta tow based approaches[81]. Feeding scars (which are generally a sign of proximate CoTS[24,79]) are also used by culling program divers to help locate individuals hidden in the reef matrix[82]. Additionally, repeated site visits (as is the case for the sites we consider here) limit the number of CoTS that go undetected at a location[12]. The detection rates described above (i.e. 19%, 82% and 82%), were

therefore applied to the CoTS population within a management site independent of coral cover or structural complexity. This was such that if an individual was observable (given size-dependent detection and population abundance biases; described below under "CoTS management intervention") then it was potentially controllable.

**CoTS reproduction formulation.** CoTS are a free-spawning invertebrate (i.e. releases their gametes directly into the water column), relying upon prevailing conditions to achieve fertilisation and transport larvae[49,50,83]. Recruitment of CoTS was distilled into two key components—self-recruitment (same sub-reef site) from reproductively mature adults and immigrants from external sources such as neighbouring sites on the same reef and/or other hydrodynamically connected reefs[28,37,49]. Due to a limited understanding of CoTS early life history and ecological processes[54], we focused only on successfully recruited and settled juveniles (age-0). Self-recruitment was represented by a novel variant of a stock-recruitment curve and immigration via a fitted constant.

Large individual CoTS contribute to the reproductive potential of a population disproportionately to their size[50,84]. Moreover, the zygote production arising from a given population decreases as the female proportion of the population decreases[50]. In our model, we therefore formulated recruitment in terms of female gonad mass, which in turn was a function of their size. To calculate the expected contributions of individuals within an age class, we used the biometric relationships of size-at-age[75,85,86] and gonad weight-at-size[84]. Given the spatially broader and contemporary relevance of MacNeil et al.[75] to our dataset (central Great Barrier Reef; years 2013–2018) relative to the earlier works of Lucas[85] and Stump[86], we used the MacNeil et al.[75] parametrisation of the von Bertalanffy growth curve. We implemented their model fitted to all 17 reefs rather than any particular reef or subset, to capture mean dynamics across multiple environmental and CoTS density realisations. This was used in conjunction with the gonad weight-at-size model of Babcock et al.[84] to model gamete production arising from reproductively mature CoTS. Size-at-age suggested by MacNeil et al.[75]. is consistent with the size-age classes implemented here in our model[70,74].

CoTS reach maturity at ~2 years of age[56,87]. Consequently, to obtain the expected reproductive potential of a population, we modelled the contributions of age-2 and age-3+ female ($F$) CoTS. The expected gonad weight GW (g), of an age-2 female CoTS of size $D$ on day $d$ was modelled by:

$$\mathrm{GW}_2^\mathrm{F}(d) = 2.2846 e^{0.0116 \cdot \left(349 - (349 - 0.03) \cdot e^{-0.54 \cdot (2 + d/365)}\right)} \quad (1)$$

Defining $M^{\mathrm{CoTS}}(a)$ to be the mortality rate of an age-$a$ CoTS, we calculate the expected gonad weight of an age-3+ individual as:

$$\mathrm{GW}_{3+}^\mathrm{F}(d) = 2.2846 \cdot \left(1 - e^{-M^{\mathrm{CoTS}}(3+)}\right) \cdot \sum_{i=0}^{\infty} e^{-i \cdot M^{\mathrm{CoTS}}(3+)} \\ \left(e^{0.0116 \cdot \left(349 - (349 - 0.03) \cdot e^{-0.54 \cdot (3 + d/365 + i)}\right)}\right) \quad (2)$$

This series is convergent by the ratio test, since the limiting ratio must be the proportion of individuals that survive each day. Therefore, for mathematical tractability, we estimated the average contribution of individuals aged up to 14 years (summing from $i = 0$ to $i = 14 - 3$, where age is $3 + i$). Considering individuals up to an age of 14 years is consistent with hypothesised upper bounds of longevity in the field[31].

Our novel formulation of the Beverton-Holt stock-recruitment relationship linked reproductively mature adult females (sex ratio Sr and proportion that spawn $P_s$) to the number of successfully settled juveniles through the spawned proportion of their gonad weight (Fdy). As a scaffold, we employed a mathematically tractable Beverton-Holt formulation that described the steepness of the stock-recruitment relationship in terms of a parameter, $h_{\mathrm{sp}}$[88]. For shape parameters $\alpha$ and $\beta$, our novel modified Beverton-Holt stock-recruitment relationship describing this relationship was calculated as a function of reproductively mature CoTS of age-$a$, $N_{d-\delta_{\mathrm{sp}},a}^y$ (age-2 and age-3+), on day $d - \delta_{\mathrm{sp}}$. The relationship was given by:

$$R(d) = \frac{\alpha \cdot \sum_{a=2}^{3+} \left(\left(\mathrm{Sr} \cdot \mathrm{P_s} \cdot N_{d-\delta_{\mathrm{sp}},a}^y\right) \cdot \left(\mathrm{Fdy} \cdot \mathrm{GW}_a^\mathrm{F}\left(d - \delta_{\mathrm{sp}}\right)\right)\right)}{\beta + \sum_{a=2}^{3+} \left(\left(\mathrm{Sr} \cdot \mathrm{P_s} \cdot N_{d-\delta_{\mathrm{sp}},a}^y\right) \cdot \left(\mathrm{Fdy} \cdot \mathrm{GW}_a^\mathrm{F}\left(d - \delta_{\mathrm{sp}}\right)\right)\right)} \quad (3)$$

Where parameter $\alpha$ was the asymptotic spawn per unit biomass and $\beta$ controlled the rate at which $\alpha$ was approached as the spawning biomass increased. Here our units of biomass were the gonad weights of spawning individuals. Defining the virgin spawning biomass of age-$a$ CoTS on day $d$ to be $K_d^{\mathrm{sp}}(a)$ and the virgin recruitment to be $R_0$, we obtained the spawned gonad-weight per recruit, $\mathrm{SPR}_0$, to be

$$\mathrm{SPR}_0\left(d - \delta_{\mathrm{sp}}\right) = \sum_{a=2}^{3+} \left(\left(\mathrm{Sr} \cdot \mathrm{P_s} \cdot K_{d-\delta_{\mathrm{sp}}}^{\mathrm{sp}}(a)\right) \cdot \left(\mathrm{Fdy} \cdot \mathrm{GW}_a^\mathrm{F}\left(d - \delta_{\mathrm{sp}}\right)\right)\right)/R_0 \quad (4)$$

And therefore, shape parameters were given by:

$$\alpha = \frac{\beta + \sum_{a=2}^{3+} \left(\left(\mathrm{Sr} \cdot \mathrm{P_s} \cdot K_{d-\delta_{\mathrm{sp}}}^{\mathrm{sp}}(a)\right) \cdot \left(\mathrm{Fdy} \cdot \mathrm{GW}_a^\mathrm{F}\left(d - \delta_{\mathrm{sp}}\right)\right)\right)}{\mathrm{SPR}_0} \quad (5)$$

and

$$\beta = \frac{\left(1 - h_{\mathrm{sp}}\right) \cdot \sum_{a=2}^{3+} \left(\left(\mathrm{Sr} \cdot \mathrm{P_s} \cdot K_{d-\delta_{\mathrm{sp}}}^{\mathrm{sp}}(a)\right) \cdot \left(\mathrm{Fdy} \cdot \mathrm{GW}_a^\mathrm{F}\left(d - \delta_{\mathrm{sp}}\right)\right)\right)}{5h_{\mathrm{sp}} - 1} \quad (6)$$

The spawners per recruits, $\mathrm{SPR}_0$, was found through equilibrium calculations. Equilibrium calculation amounted to finding the number of spawners required to produce a constant population indefinitely (an equilibrium) under only CoTS natural mortality (sensu[63]). The natural mortality rate was modelled to decrease with increasing CoTS size and age such that younger CoTS had a much higher natural mortality rate than older individuals[37]. For a basal mortality rate of $\omega$ and an age-dependent decay rate of $\lambda$ the mortality rate of an age-$a$ CoTS[37] was given by:

$$M^{\mathrm{CoTS}}(a) = \omega + \frac{\lambda}{a + 1} \quad (7)$$

For $K_d^{\mathrm{sp}}(a)$ individuals in the virgin population of age-$a$ on day $d$ and for the corresponding daily instantaneous mortality of an age-$a$ individuals, $M^{\mathrm{CoTS}}(a)$, we calculated the number of CoTS in each age class recursively. For a virgin population recruitment level of $R_0$, the number of CoTS in their first year in said population on day $d$ was described by:

$$K_d^{\mathrm{sp}}(0) = R_0 \cdot e^{-d \cdot M^{\mathrm{CoTS}}(0)} \quad (8)$$

For age classes $a \in \{1, 2\}$:

$$K_d^{\mathrm{sp}}(a) = R_0 \cdot e^{-\left(365 \cdot \sum_{i=0}^{a-1} M^{\mathrm{CoTS}}(i) + d \cdot M^{\mathrm{CoTS}}(a)\right)} \quad (9)$$

And the absorbing age class:

$$K_d^{\mathrm{sp}}(3+) = \frac{R_0 \cdot e^{-\left(365 \cdot \sum_{i=0}^{2} M^{\mathrm{CoTS}}(i) + d \cdot M^{\mathrm{CoTS}}(3+)\right)}}{1 - e^{-M^{\mathrm{CoTS}}(3+)}} \quad (10)$$

Hence, we obtained an expression for the equilibrium spawned gonad weight per recruit as a function of each age classes' natural mortality:

$$\mathrm{SPR}_0(d) = (\mathrm{Sr} \cdot \mathrm{P_s} \cdot \mathrm{Fdy}) \cdot \\ \left(\mathrm{GW}_2^\mathrm{F}(d) \cdot e^{-\left(365 \cdot \sum_{i=0}^{1} M^{\mathrm{CoTS}}(i) + d \cdot M^{\mathrm{CoTS}}(2)\right)} + \mathrm{GW}_{3+}^\mathrm{F}(d) \cdot \frac{e^{-\left(365 \cdot \sum_{i=0}^{2} M^{\mathrm{CoTS}}(i) + d \cdot M^{\mathrm{CoTS}}(3+)\right)}}{1 - e^{-M^{\mathrm{CoTS}}(3+)}}\right) \quad (11)$$

**CoTS management intervention.** A range of factors influences CoTS management efficacy and the number of CoTS that can be removed however a key driver of heterogenous impacts are CoTS size and age as well as population density[12,31,63]. We incorporated a formally fitted hyperstability relationship to represent management intervention[63]. This was based on an empirically derived relationship using data collected by Fisk and Power[89] which comprised simultaneous CPUE (CoTS.h$^{-1}$) and CoTS density estimates (CoTS.ha$^{-1}$) over a period of 20 weeks during 1995–1996. A recent study by MacNeil, et al.[76] inter-calibrated multiple data sources (including mark-recapture data) and found that Australian Marine Park Tourism Operator (AMPTO) CPUE observations (to which we fit our model) to be informative of reef-scale CoTS densities. Both MacNeil et al.[76] and Plagányi et al.[63] found the relationship between CPUE and known CoTS density to be hyperstable; we used the empirically validated relationship of Plagányi et al.[63] as it also comprised smaller CoTS compared to MacNeil et al.[76]. Here, during year $y$ and on day $d$, CoTS were extracted through the term $\phi_a^{\mathrm{CoTS}}\left(\mathrm{Fp}_d^y \cdot N_{d,a}^y\right)$ where $\mathrm{Fp}_d^y$ was the fished proportion of individuals captured with $\phi_a^{\mathrm{CoTS}}$ selectivity for age-$a$ individuals given $N_{d,a}^y$ CoTS of age-$a$, $t_d^y$ minutes of diver effort and a catch-per-unit-effort rate of $\mathrm{CPUE}_d^y$. The fished proportion of the sub-reef CoTS population was the number of individuals caught relative to the available or accessible population:

$$\mathrm{Fp}_d^y = \frac{\mathrm{CPUE}_d^y \cdot t_d^y}{\sum_a \phi_a^{\mathrm{CoTS}} N_{d,a}^y} \quad (12)$$

For a given catchability, $q$, and hyperstability parameter, $h$, catch-per-unit-effort was computed by:

$$\mathrm{CPUE}_d^y = q \left(\sum_a \phi_a^{\mathrm{CoTS}} N_{d,a}^y\right)^h \quad (13)$$

Here the catchability accounted for the probability of removing a CoTS given its detected and the hyperstability parameter characterised the rate at which the removal rate saturated. The removal rate decoupled (saturated) when controlling high densities of CoTS populations due to handling time constraints[63]. Hence the number of CoTS removed on a given voyage to a reef zone, $N_{d,a}^{y,\mathrm{culled}}$, was described by:

$$N_{d,a}^{y,\mathrm{culled}} = \phi_a^{\mathrm{CoTS}} N_{d,a}^y \left(\frac{\mathrm{CPUE}_d^y \cdot t_d^y}{\sum_a \phi_a^{\mathrm{CoTS}} N_{d,a}^y}\right) \quad (14)$$

Explicitly capturing management efficacy allowed us to resolve and evaluate management's impacts on CoTS population and coral abundance dynamics given different environmental perturbations.

**CoTS population dynamics.** The cumulative population dynamics equation for CoTS was obtained by amalgamating reproduction and mortality sources. We denoted the annual immigration of successful settling CoTS into a sub-reef zone on day $d$ by $I_d^{\text{CoTS}}$, with recruitment variability (of both immigration and self-recruitment) modelled by the parameter $r_{y,\text{reef}}^{\text{rec}}$ with an associated standard deviation of $\sigma_R$. Whether recruits settled into a sub-reef zone on day $d$ was captured through an impulse function $\delta_1\!\left[d + 1 - \delta_{\text{sp}}\right]$. The impulse function took a value of one if its argument was true (an argument of zero i.e. $d + 1 - \delta_{\text{sp}} = 0$) and zero if the argument was false (a non-zero argument i.e. $d + 1 - \delta_{\text{sp}} \neq 0$) which was a function of CoTS pelagic duration, $\delta_{\text{sp}}$. That is, all settlement was assumed to occur each year one day after the pelagic duration had elapsed. The population dynamic equations during year $y$ of age-0 CoTS, $N_{d,a}^y$, was therefore expressed by:

$$N_{d+1,0}^y = N_{d,0}^y e^{-M^{\text{CoTS}}(0)} + \left(R\left(N_{365,2}^{y-1} + N_{365,3^+}^{y-1}\right) + I_{d+1}^{\text{CoTS}}\right)e^{r_{y,\text{reef}}^{\text{rec}} - \sigma_R^2/2}\delta_1\left[d+1-\delta_{\text{sp}}\right] \quad (15)$$

For CoTS aged 1+ years, we omitted the recruitment term and incorporated CoTS lost due to interactions with the management control program, $N_{d,a}^{y,\text{culled}}$. We also captured negative feedback of low prey availability upon survival $f\!\left(C_{y,d}^{\text{f}}\right)$ where $C_{y,d}^{\text{f}}$ was the availability of preferred coral prey (sensu[37]). This was such that the mortality rate increased as preferred prey decreased. The population dynamics of CoTS aged over 1 year (where $a$ references CoTS age) was:

$$N_{d+1,a}^y = N_{d,a}^y e^{-f\left(C_{y,d}^f\right)M^{\text{CoTS}}(a)} - N_{d,a}^{y,\text{culled}} \quad (16)$$

Boundary conditions ensured that the respective age classes incremented to the next age class at the end of each year. The exceptions were that there were initially no age-0 individuals at the start of the year and that age-3+ CoTS incremented back into the age-3+ class due to it being an absorbing state. Whilst Eq. (15) and Eq. (16) describe the basic population dynamics of CoTS, these dynamics were subject to perturbation through interspecific interactions with corals (see below) as well as management interventions and therefore, indirectly by thermal bleaching and cyclone events.

**Coral population dynamics.** *Acanthaster spp.* exhibit strong prey preferences and frequently consume preferred coral taxa disproportionately to their abundance[22,24,31]. Reciprocally, low preferred coral abundance can induce deterioration in the condition of CoTS and increase their mortality[31,37]. Resolving the differential impact of CoTS consumption upon coral taxa and the reciprocal impact of low preferred prey availability are incorporated to quantify and characterise the efficacy of management intervention in terms of coral cover dynamics. To ease subsequent notation, we have omitted superscripts that reference fast-growing corals, f, and slow-growing corals, s in our following parameter descriptions.

Cyclones and bleaching are two principle threats to the Great Barrier Reef in addition to CoTS predation[30]. In our model, both coral groups are negatively impacted by cyclones, $M_{y,d}^{\text{Cyc}}$, and bleaching, $M_{y,d}^{\text{Ble}}$, through the removal of coral biomass. Cyclones, as treated as an impulse source of mortality[28] for reefs within an explicit spatial subregion of influence and bleaching, is computed regionally over a period of 4 months between January and April[43,90,91] based on accumulated thermal stress. Bleaching and cyclone-induced mortalities are expressed as proportions. This is such that the proportion of corals that survive a given perturbation on day $d$ is $1 - M_{y,d}^{\text{Cyc}}$ for a cyclone event or $1 - M_{y,d}^{\text{Ble}}$ in the case of bleaching-induced mortality. Implementation of coral dynamics involves computing coral growth minus loss due to CoTS predation, $Q_{y,d}$, followed by abiotic factors—cyclones, $M_{y,d}^{\text{Cyc}}$, and bleaching, $M_{y,d}^{\text{Ble}}$. We modelled joint coral cover dynamics of the fast-growing coral group, $C_{y,d}^{\text{f}}$, through:

$$C_{y,d+1}^f = C_{y,d}^f\left(1 + r^f\left(1 - \frac{C_{y,d}^f + C_{y,d}^s}{K^{\text{coral}}}\right) - Q_{y,d}^f - M_{y,d}^{f,\text{Ble}} - M_{y,d}^{f,\text{Cyc}}\right) \quad (17)$$

Similarly, the dynamics of the slow-growing coral group, $C_{y,d}^{\text{s}}$, were given by:

$$C_{y,d+1}^s = C_{y,d}^s\left(1 + r^s\left(1 - \frac{C_{y,d}^f + C_{y,d}^s}{K^{\text{coral}}}\right) - Q_{y,d}^s - M_{y,d}^{s,\text{Ble}} - M_{y,d}^{s,\text{Cyc}}\right) \quad (18)$$

The distinguishing features between these two groups are their growth rates, $r$, their susceptibility to CoTS predation, $Q_{y,d}$, and their susceptibility to thermal stress, $M_{y,d}^{\text{Ble}}$, and cyclones, $M_{y,d}^{\text{Cyc}}$. We implemented the formally fitted coral growth rates of Morello et al[37].

**CoTS-coral consumption.** As preferred taxa are depleted, consumption increasingly switches to non-preferred taxa and individuals increase searching behaviour[22,24]. We captured this dynamic such that as preferred prey items, $C_{y,d}^{\text{f}}$, are depleted relative to the joint carrying capacity, individuals switched to consumption of non-preferred prey items, $C_{y,d}^{\text{s}}$. We hereafter refer to this as a prey switching function which we defined on day $d$ of year $y$ to be $\rho_{y,d}$ through which we represented predation pressure on $C_{y,d}^{\text{f}}$. Predation pressure on $C_{y,d}^{\text{s}}$ was modelled

through its reciprocal, $1 - \rho_{y,d}$. Moreover, the consumption rate and feeding efficiencies of CoTS were modelled to be subject to density dependence at low abundance levels (sensu[37]). This was such that density dependence was incorporated through a Holling type II interaction form where the CoTS per capita consumption rate was $0.5p_1^{\text{f}}$ at low population abundance and was asymptotic to $p_1^{\text{f}}$. The rate at which the per capita consumption rate increased as a function of CoTS abundance depending on the parameter $p_2^{\text{f}}$ with f denoting reference to $C_{y,d}^{\text{f}}$ (trading superscripts as above for $C_{y,d}^{\text{s}}$). The foraging efficiency of CoTS was incorporated through a multiplicative prey switching term. We developed our prey switching as a logistics function of preferred prey depletion relative to the joint coral carrying capacity of fast and slow-growing corals, $K^{\text{coral}}$, on a reef. The term was given by

$$\rho_{y,d} = \frac{1}{1 + e^{-70\left(C_{y,d}^f/K^{\text{coral}} - 0.10\right)}} \quad (19)$$

Thus, the density-dependent total consumption of $C_{y,d}^{\text{f}}$ within a reef site for a population of CoTS and $C^{\text{f}}$ depletion level was modelled as:

$$Q_{y,d}^f = \rho_{y,d} \cdot \frac{p_1^f}{1 + e^{-\sum_{a=1}^{3^+} N_{d,a}^y/p_2^f}} \cdot \textstyle\sum_{a=1}^{3^+} N_{d,a}^y \quad (20)$$

And as feeding efficiency upon preferred prey items reduced, feeding rates upon non-preferred prey items, $C_{y,d}^{\text{s}}$, increased as described by:

$$Q_{y,d}^s = \left(1 - \rho_{y,d}\right) \cdot \frac{p_1^s}{1 + e^{-\sum_{a=1}^{3^+} N_{d,a}^y/p_2^s}} \cdot \textstyle\sum_{a=1}^{3^+} N_{d,a}^y \quad (21)$$

The effects of food limitation were modelled through scaling CoTS mortality rates as a function of food availability. The relationship between preferred prey abundance and CoTS mortality incorporated the prey switching term[37] as per:

$$f\left(C_{y,d}^f\right) = 1 - \widetilde{p} \cdot \rho_{y,d} \quad (22)$$

The coral-induced mortality was tuned to the management control data available for CoTS and comparable to results obtained via estimates based on the Long Term Monitoring Program data for Lizard Island 1994–2011[37].

**Thermal stress and coral mortality.** Bleaching-induced coral mortality is a principal threat to coral reefs[3,30,92,93]. Bleaching-induced mortality was formulated as a sigmoidal function relating accumulated thermal stress (Degree Heating Weeks[94]) to coral mortality based on the relationship described by Condie et al.[17]. The sigmoidal relation described by Condie, et al[17]. is parametrised to the 2016 mass coral bleaching event on the Great Barrier Reef[92]. Here we extended their bleaching-induced mortality formulation to a daily sub-reef scale and considered interactions with co-occurring cyclones. The increased complexity of our formulation provided scope for capturing differential susceptibilities of corals under management intervention between and within reefs based on different disturbance regimes and their footprint/s.

The approach we implemented was parametrised to family taxonomic resolution[17] which we aggregated into a bleaching-induced mortality response for CoTS' preferred fast-growing corals and non-preferred slower-growing corals. For a given level of thermal stress in terms of Degree Heating Weeks, DHW, the bleaching-induced mortality was calculated over a defined bleaching period, $T^{\text{Blea}}$. The bleaching period was assumed here to last from the 1 January to the 30 of April each year (119 days) consistent with the austral summer and both the 2016 and 2017 mass bleaching events on the Great Barrier Reef[72,90,91]. To account for differences in the coral groups' susceptibility to bleaching, we defined parameter $T_y^{\text{g}}$ as the susceptibility of group g to bleaching-induced mortality in a particular year. Where bleaching cooccurred with tropical cyclone/s, the DHW value was reduced to reflect the cooling induced by tropical cyclone/s at relevant locations leading up to and during the bleaching period (cyclone interactions discussed below). For purposes of induced cooling, the days since the last tropical cyclone are reset each season. The proportion of initial coral cover lost over the bleaching period given DHW accumulated thermal stress and $d_{\text{postTC}}$ days post-cyclone event was described by:

$$M_y^{\text{g,Blea}}\left(\text{DHW}, d_{\text{postTC}}\right) = 1 - e^{-0.01 \cdot \exp\left(\text{DHW} - T_y^g - \text{DHW}_{\text{cool}}(d_{\text{postTC}})\right)} \quad (23)$$

With superscript g referencing either fast- or slow-growing coral groups via f or s respectively. The annual rate was calculated each day to accommodate daily changes in $\text{DHW}_{\text{cool}}$ as the cooling signal decayed. The daily rate was then computed from the dynamic annual rate. To obtain the daily rate of loss required factoring in that $M_y^{\text{g,Blea}}$ was the culmination of the compounding daily proportional losses. This necessitated the use of the geometric mean to approximate the daily bleaching mortality to avoid underestimating mortality. Hence average daily loss due to bleaching was computed by:

$$M_{y,d}^{\text{g,Blea}}\left(\text{DHW}, d_{\text{postTC}}\right) = 1 - \left(1 - M_y^{\text{g,Blea}}\left(\text{DHW}, d_{\text{postTC}}\right)\right)^{1/T^{\text{Blea}}} \quad (24)$$

Coral susceptibility to bleaching-induced mortality was modelled through a group-specific 'tolerance' based on the coral group's intrinsic growth rate

parameter[17]. This was such that faster-growing branching and tabular corals were more susceptible to thermal stress than their massive slower-growing counterparts[48,95,96]. The thermal tolerance of coral group g described in terms of its intrinsic growth rate, $r^g$, was modelled through[17]:

$$T_0^g = 3.5 - 5 \cdot 365 \cdot r^g \tag{25}$$

Where g referenced either fast- or slow-growing coral groups as previously. In addition to group-specific thermal tolerances, we also captured tolerance dynamics in response to prior bleaching events.

Recent back-to-back bleaching events on the Great Barrier Reef have highlighted that coral communities were less severely impacted if they had experienced significant bleaching in the previous year[43]. Moreover, their adaptation and/or acclimation capacity is related to the magnitude of their prior exposure[43]. Thus, to plausibly capture the cumulative impact of repeated bleaching events it was necessary to model the adaptation and/or acclimation of corals contingent on their recent thermal history. This was recursively achieved through incorporating a sub-reef zone's bleaching history into the thermal tolerance of its constituents[17]. With superscript g referencing fast- or slow-growing corals as before, we represented the adaptation and/or acclimation of each group as per Condie et al.[17] through a parameter A. This was such that a higher value of A corresponded to greater reductions in the sensitivity of corals to thermal stress. Three values were considered representative of a moderate response (A = 5), low response (A = 2.5) and no response (A = 0). We defined the overall bleaching-induced mortality from the previous year as $M_y^{Blea}$ (inclusive of accumulated thermal stress and cyclone interactions). The tolerance of group g was therefore described by:

$$T_{y+1}^g = T_y^g (1 + A)^{M_y^{Blea}} \tag{26}$$

This formulation allowed thermal tolerance to develop proportional to bleaching mortality and adaptation and/or acclimation capacity of corals[17]. Combing the thermal tolerance of corals into Eq. (24) resulted in the computation of the average daily loss due to bleaching over a reef site as:

$$M_{y,d}^{g,Blea}\left(DHW, d_{postTC}\right) = 1 - e^{-0.01/T^{Blea} \cdot \exp\left(DHW - T_{y-1}^g(1+A)^{M_{y-1}^{g,Blea}} - DHW_{cool}\left(d_{postTC}\right)\right)} \tag{27}$$

To incorporate potential cyclonic cooling signals into the adaptive capacity of coral for the following year, we recalculated $M_y^{Blea}$ for use in the following years adaptive capacity (Eq. 26) as:

$$M_y^{Blea} = 1 - \prod_{i=1}^{T^{Blea}}\left(1 - M_{y,i}^{g,Blea}\left(DHW, d_{postTC}\right)\right) \tag{28}$$

We compared projection scenarios both with and without coral adaptive capacity to quantify the cumulative impact of increased bleaching severity upon coral communities under management intervention. For projections, a single thermal stress event was specified in the year 2022 to evaluate the consequence of corallivore management in its aftermath—the focus of the present study. Thermal stress events were not stochastic as per cyclone events to establish the efficacy of CoTS management (the output) in response to a bleaching event (the input) given the expected influence of typical cyclone regimes. The year 2022 was selected to simulate the thermal stress event as this was approximately halfway through our projection period (2018–2029) and allowed sufficient time to evaluate the interaction between corallivore management and the immediate and lasting impacts of thermal stress events.

The lasting potential impacts of coral bleaching encompass reduced growth rates as well as reductions in the reproductive output of corals and lower recruitment success[97,98]. Bleaching may physiologically compromise coral energy reserves and reduce growth[6,97,99]. Reduced growth following bleaching is typically reported to be between one and four years, though may be longer[45]. Additionally bleaching may result in the collapse of coral stock-recruitment dynamics (larval production and recruitment success) due to reductions in local and regional adult broodstock and large areas of poor settlement substrates for larvae (unstable dead corals)[43,98]. For example, the loss of adult broodstock can reduce larval production to as little as 10% of historical averages[97,98]. Cumulatively, reduced growth and a breakdown of stock-recruitment dynamics are likely to delay the recovery of coral assemblages[43]. Here we captured the outcome of these processes through tuning a reduction in the net coral growth rate (previously described parameters $r^f$ and $r^s$) for a period of three years following a thermal stress event. This was such that net growth was reduced by half over the three years. This captured the net outcome of lower growth rates and reproductive output of surviving corals, lower settlement success and/or management sites receiving fewer recruits from other locations.

**Tropical cyclone events**. The approach we implemented here builds on that of Condie et al.[28] in that cyclone intensity was assumed to be empirically based[100] with rendered coral mortality random within intensity-dependent bounds[28]. We extend this basic model by resolving cyclone intensity contours as a function of cyclone centre displacement and by scaling up impact footprints to reflect empirical cyclone sizes since 1985 on the Great Barrier Reef[101]. Our augmented approach employed cyclone intensity (implicitly through maximum wind speed) as a proxy for wave energy and explicitly resolved cyclone size and how likely coral mortality varied as a function of cyclone size, intensity and reef displacement. We

assumed a cyclone impacts a reef if there existed any spatial overlap between the reef and the cyclone with coral damage rendered precautionarily based on the bounds prescribed by the highest cyclone intensity experienced by the reef. Wind speed and duration are determinant factors of wave energy[102], but alternative work suggests that maximum wind speed is the best predictor of coral damage with duration or cumulative wind energy offering only marginal improvement in the prediction of coral damage[100]. Additionally, coral community structure may lead to different susceptibilities to cyclone-induced damage which may affect damage patterns[103]. Given that MICE restrict their focus to key system dynamics only, we therefore used cyclone intensity and size and different intensity-damage relationships for fast and slow-growing coral groups. This was such that increased tropical cyclone intensity increased damage ranges for corals[28] with wind speeds (intensity) decaying spatially away from the radius of maximum wind speed[104]. This regionally contextualised our local-scale model under cyclone perturbation/s to characterise potential destructiveness owing to their size and the distribution of reefs[105].

Cyclones were modelled to inflict damage to corals on a reef as an impulse[28,100] with their frequency modelled stochastically via a Poisson process[106]. Here, cyclone events occurred with a mean daily arrival rate of $\lambda$[106] between the 1 November and the 1 April. For simplicity, the damaged region of a cyclone was modelled as a circular footprint with wind speed intensity contours. The centres of such footprints were calculated by uniformly generating an angular and radial displacement of the cyclone centre from the centre of a circular model focal region (diameter 300 km). This was such that a cyclone could be centred external to the focal area if the peripheral damage region overlapped the focal area. A circular model focal region capturing overlapped footprints of externally centred cyclones is consistent with the empirical work upon which the arrival rate was computed (hexagonal;[106]).

For an event rate of $\lambda^{cyc}$ and an observation period of t, the probability of observing a cyclone impacting the model region was given by:

$$Pr(N \geq 1) = 1 - e^{-\lambda^{cyc}t} \tag{29}$$

where the observation period was fixed at the model temporal resolution, t = 1 (days), and the cyclone event rate was calculated from the annual event rate for the region[106] and restricted to the period over 1 November to 1 April (152 days). This yielded $\lambda^{cyc} \sim 0.0026$. To obtain events we sampled $Pr(N \geq 1) \sim U(0, 1)$ and found the time of the next event via:

$$t = \frac{-1}{\lambda^{cyc}}\ln(1 - Pr(N \geq 1)) \tag{30}$$

If the arrival time of the next cyclone event was within a day then a cyclone impulse was simulated to occur, otherwise no cyclone impulse occurred. This approach was repeated for each day within the cyclone season and implicitly—and realistically—assumed a maximum of 1 cyclone impulse per day with a single impulse sufficient to capture cumulative mortality.

Wind speed was assumed to be constant at a given reef such that when a reef was struck by a tropical cyclone the cyclone category was constant across all sites at said reef. However, damage to reef sites varied stochastically in conjunction with its coral community composition and its associated susceptibility (Supplementary Table 4).

The cyclone category experienced by a reef was calculated as a function of the maximum wind velocity, cyclone radius and the distance of the cyclone from the reef[104]. A reef strike was defined here as a reef's exposure to wind speeds ≥17 m.s⁻¹ (category 1 or higher, see Supplementary Table 4) associated with a tropical cyclone. The radius of maximum wind speed, $d_m$ (km), was calculated via a regression model[104] through:

$$d_m\left(V_m\right) = 39.9116075 - 0.1578765V_m - \delta\left(V_m \leq 32.5\right)\left(2.0853134\left(V_m - 32.5\right)\right) \tag{31}$$

Where $\delta\left(V_m \leq 32.5\right)$ is an impulse function taking the value of one if its argument is true (i.e. $V_m \leq 32.5$) and zero otherwise (i.e. $V_m > 32.5$). This induced a slope change in the relationship between maximum wind velocity and its radius at a wind velocity of 32.5 m.s⁻¹ (≥ category 3 intensities). However, whilst maximum wind velocity was modelled to determine $d_m$, the overall size of the cyclone was uncorrelated with its intensity. The overall size was uniformly sampled from 130 to 460 km diameter which allowed for the potential of complete focal area coverage and for a range of intensity-size relationships to be captured. Given a cyclone footprint of radius $d_0$ (km), wind velocity, V (m.s⁻¹), at a distance, d (km), was interpolated[104] through:

$$V(d) = \begin{cases} V_0 + (V_m - V_0)\left(\frac{\sqrt{d_0}-\sqrt{d}}{\sqrt{d_0}-\sqrt{d_m}}\right)^\alpha, & d \geq d_m \\ V_m, & d < d_m \end{cases} \tag{32}$$

The distance from the cyclone centre to the reef perimeter, D (km), is calculated through:

$$D = \sqrt{\left(x_{rf} - r_1 - x_{cyc}\right)^2 + \left(x_{rf} - r_1 - y_{cyc}\right)^2} \tag{33}$$

Thus, given a reef strike occurs ($\sqrt{d_0} - \sqrt{d} \geq 0$ required from non-integer $\alpha$), the wind velocity experienced at said reef due to the tropical cyclone was calculated

as $V(D)$. Wind velocity was subsequently categorised and damage to reef zone corals calculated as per Supplementary Table 4.

We resolved stochasticity in cyclone dynamics in projection scenarios. In projected scenarios cyclone arrivals, locations and intensities were probabilistically sampled and their inflicted damage upon coral communities sampled from damage ranges. Cyclone locations, their footprints, intensity ranges and corresponding damage ranges were sampled from uniform distributions. Cyclone arrivals were sampled from a Poisson distribution and considered in scenarios from 2018 to 2029. Projections were averaged over 80 simulations to capture mean dynamics and bound trajectory uncertainty due to said stochasticity.

Our cyclone model was calibrated to parameters sourced from the literature (Supplementary Tables 4-5). This was necessary since our data time series did not encompass a cyclone event and/or impacts upon a reef and cyclone-induced mortality is typically a key coral mortality source[30]. Consequently, we were unable to validate the impacts of cyclones through formal estimation in our model. However, our endeavours to source parameters from empirical and modelling studies in conjunction with our formulation allowed us to plausibly capture the cumulative outcomes of a cyclone event at discrete locations. Our cyclone model offers a limited complexity approach that is empirically grounded to simply resolve cyclone impacts in local-scale models without the need to be coupled to a regional-scale model.

### Cyclones, induced thermal stress and tactical management.
The occurrence of cyclone events was modelled to directly interact with both management interventions and thermal stress events. Cyclones were assumed to realistically preclude co-occurring co-located management interventions. This was such that a management site control visit was abandoned if a cyclone preceded or was forecast within five days of a control voyage. The later interaction of cyclones with thermal stress events operated through an induced thermal cooling of sea surface temperatures (SST) at impacted locations.

In the case of the overlapping cyclone and thermally induced bleaching events, we first accounted for cyclone impacts. This was because, in addition to physical damage to corals, cyclones have the potential for regional-scale cooling of SST which can reduce coral bleaching[43,107]. To capture this interaction, we resolved the duration[108,109] and amplitude[107] of tropical cyclone-induced cooling. We captured this interaction through Degree Heating Weeks (DHW) which is a useful metric for the accumulated thermal stress experienced by corals[94].

The duration of tropical cyclone-induced cooling was modelled through a temporal-SST response curve consistent with the work of Lloyd and Vecchi[108] and Vincent et al.[109]. Cooling rapidly occurs once a tropical cyclone arrives at a location and decays in an asymptotic manner over a period of ~40–60 days[108,109]. Temperatures however do not return to pre-cyclone levels and plateau at ~1/4 of the cooling signal amplitude below pre-cyclone levels[108,109]. We expressed this cooling response curve as it related to bleaching-induced coral mortality through DHWs.

We based the average expected DHW cooling signal on the work of Carrigan and Puotinen[107]. This was achieved through scaling the difference in amplitude of overlapping thermal stress-tropical cyclone events and thermal stress only events— a cooling signal amplitude of $DHW_{Amp} \sim 1.5$ DHW. Consistent with the model of Carrigan and Puotinen[107], we then resolved cooling within the radius of gale-force winds (category 1, $17\,\text{m.s}^{-1}$) to model tropical cyclone-induced cooling. Depending on the size of the tropical cyclone, this meant that an individual cyclone would not necessarily cool all reefs within the model region. However, the culmination of multiple cyclones may have limited bleaching exposure for corals across the region[107].

We did not treat the cooling consequences of multiple cyclones additively nor the complex interplay of oceanic feedbacks upon cyclone intensity and cooling. Such processes were beyond the scope of our study and model. If multiple cyclones occurred within our model, then the cooling signal timeline was re-initialised at impacted reefs for the last tropical cyclone at said location. Non-impacted reefs maintained the timeline for the decay of the cooling signal originating from their previous tropical cyclone interaction.

Once a tropical cyclone impacted a reef, the duration of the induced cooling signal was modelled. Price et al.[110] found that cooling decays exponentially which is reflective of the recovery of SST following tropical cyclones as demonstrated by Lloyd and Vecchi[108] and Vincent et al.[109]. We operationalised the exponential functional form in conjunction with the decay timelines of Lloyd and Vecchi[108] and Vincent et al.[109] and the DHW amplitude of Carrigan and Puotinen[107]. We modelled the level of cooling $DHW_{cool}$ after $d_{postTC}$ days post-cyclone event by:

$$DHW_{cool}\left(d_{postTC}\right) = \frac{1}{4}DHW_{Amp} + \frac{\frac{3}{4}DHW_{Amp}}{e^{d_{postTC}/10}} \tag{34}$$

This ensured that once a reef experienced a tropical cyclone event, the cooling signal initialised at $DHW_{Amp}$ and decayed to $\sim \frac{1}{4}DHW_{Amp}$ after 40–60 days[108,109]. The rate of decay was given by the e-folding time (days required for the cooling signal to be reduced by a factor of $e$) which we took to be 10. This is consistent with the results of Price et al.[110], Lloyd and Vecchi[108] and Vincent et al.[109] who found e-folding times ranging from 5 through to 20 days. Thermally induced bleaching mortality of corals was computed after cyclone physical damage and cooling had been accounted for.

### Formal model fitting.
We formally fitted our coral-CoTS model simultaneously to coral cover data, catch-per-unit-effort data and catch numbers obtained from the management control program with dive effort (minutes) treated as an input (visits summarised in Supplementary Table 7)[12]. Simultaneously fitting CoTS and coral dynamics at concurrent locations was useful here as it allowed for coral cover trajectories to help inform local CoTS abundance (sensu CoTS feeding vs. coral trajectories[63,79] and local site fidelity[24]). Our model also used Long Term Monitoring Program (LTMP) data (based on manta tows and provided by the Australian Institute of Marine Science) which provides an independent index of relative abundance of CoTS. This was such that our model here was developed and parametrised based on an earlier version[37,111] which did not use CPUE information but was fitted to the LTMP data on CoTS relative abundance, as well as the corresponding coral cover, to estimate a number of CoTS-coral interaction parameters used in the present model (Supplementary Table 3).

Fitting and estimation of our model were achieved through Maximum Likelihood Estimation (MLE). Our objective function was the outcome of combining the negative log-likelihood contributions arising from fitting the model to multiple sets of location-specific data, across a range of environmental and ecological realisations, in conjunction with penalty terms. Specifically, we fitted coral cover (data series $x^{Coral}$) and CoTS CPUEs (data series $x^{CoTS}$) at each management site which contained $n_{Coral}$ and $n_{CoTS}$ data points respectively. This involved fitting parameters that were specific to management sites (e.g. thermal stress - DHW), reefs (e.g. recruitment variability) as well as those that were common amongst reefs (e.g. CoTS consumption rates). A parametrisation that optimised one contribution was unlikely to optimise all contributions and hence we obtained a parametrisation across all reefs and sub-regions. For a modelled catch of $N$ (sum of catches across age classes), a catchability coefficient (a constant of proportionality) of $q_{LL}^{prop}$, and data standard deviation of $\sigma_{LL}$ our likelihood contribution arising from a management site CPUEs was given by:

$$-\log L\left(q_{LL}^{prop}N, \sigma_{LL}{}^2|x_i^{CoTS}\right) = n_{CoTS}\ln(\sigma_{LL}) + \sum_{i=1}^{n_{CoTS}}\frac{\left(\ln(x_i^{CoTS}) - \ln(q_{LL}^{prop}N_i)\right)^2}{2\sigma_{LL}{}^2} \tag{35}$$

From which the data series variance and catchability coefficient were computed for the maximum likelihood estimate. The derived variance and the catchability were respectively computed as per:

$$\sigma_{LL} = \sqrt{\frac{1}{n_{CoTS}}\sum_{i=1}^{n_{CoTS}}\left(\ln(x_i^{CoTS}) - \ln(q_{LL}^{prop})\right)^2} \tag{36}$$

and

$$q_{LL}^{prop} = \frac{1}{n_{CoTS}}\sum_{i=1}^{n_{CoTS}}\left(\ln(x_i^{CoTS}) - \ln(N_i)\right) \tag{37}$$

Similarly, the likelihood contribution arising from fitting to a management site coral cover with standard deviation $\sigma_{Coral}$ was described by:

$$-\log L\left(\frac{C_{y,d}^f + C_{y,d}^s}{K^{coral}}, \sigma_{Coral}{}^2|x_i^{Coral}\right) = n_{Coral}\ln(\sigma_{Coral}) + \sum_{i=1}^{n_{Coral}}\frac{\left(\ln(x_i^{Coral}) - \ln\left(\frac{C_{y,d}^{f,i} + C_{y,d}^{s,i}}{K^{coral}}\right)\right)^2}{2\sigma_{Coral}{}^2} \tag{38}$$

Where the standard deviation was given by:

$$\sigma_{Coral} = \sqrt{\frac{1}{n_{Coral}}\sum_{i=1}^{n_{Coral}}\left(\ln(x_i^{Coral}) - \ln\left(\frac{C_{y,d}^{f,i} + C_{y,d}^{s,i}}{K^{coral}}\right)\right)^2} \tag{39}$$

We computed the negative log-likelihood objective function by summing the contributions from all management sites across considered reefs.

Fitting was conducted through the modelling language Automatic Differentiation Model Builder (ADMB) which implements a Quasi-Newton optimisation algorithm for estimation of parameters and provides Hessian based estimation of standard errors[112]. Penalty terms were added to our likelihood function to integrate a prior understanding of system dynamics and to reduce model variability. Penalty terms encompassed recruitment variability and the magnitude of catches observed in the data.

Recruitment was expressed through recruitment deviations, $r_y$, given a standard deviation of $\sigma_R$ about underlying modelled recruitment (sum of self-recruitment and immigration sources described previously). The recruitment variability negative log-likelihood penalty contribution was given by:

$$-\log L\left(0, \sigma_R^2|r^{rec}\right) = \sum_{y=1}^{\#Years}\sum_{reef=1}^{\#Reefs}r_{y,reef}^{rec}/2\sigma_R^2 \tag{40}$$

An additional penalty term for model deviations from the magnitude of observed catches was encompassed. This was such that a constant of proportionality relating modelled catches to observed catches tended to one. For an allowed standard deviation of $\sigma_{CM}$, the likelihood function was penalised for deviations from unity proportionality, $r^{CM}$, through:

$$-\log L\left(0, \sigma_{CM}^2|r^{CM}\right) = \sum_{zone=1}^{\#Zones}r_{zone}^{CM}/2\sigma_{CM}^2 \tag{41}$$

Model simulations were conducted in ADMB with output analysis and visualisation conducted in MATLAB.

**Sensitivity to CoTS control**. To test whether our projected scenarios were consistent with the period over which data were collected, we conducted a model-based before and after comparison to the impact of control. Specifically, we used the fitted trajectory for sites, including both the coral data and CoTS control data (voyages and time spent), and compared this to the model-suggested coral trajectories if CoTS control had not taken place. These were modelled over the fitted period (2013–2018) and, unlike the projected scenarios (2019–2029), were variable in terms of the timing of control (amount of time between visits was variable), the amount of time spent at sites (not a consistent number of dive minutes per visit), CoTS dynamics (recruitment was fitted and hence different annually and between reefs), and in the level of thermal stress they experienced (different sites experienced different effective levels and some sites experience back-to-back events).

**Reporting summary**. Further information on research design is available in the Nature Research Reporting Summary linked to this article.

## Data availability
The raw data are not available as we are not the data custodian. For queries about data detailing the Crown-of-thorns Starfish Control Program on the Great Barrier Reef, please email J.G.D.R. which will be redirected to the data custodian, the Great Barrier Reef Marine Park Authority. The Crown-of-Thorns Starfish Control Program data can be requested via contacting the Great Barrier Reef Marine Park Authority via either emailing info@gbrmpa.gov.au or via filling out the form at www.gbrmpa.gov.au/about-us/contact-us. Source data (model outputs for main figures) are provided with this paper.

## Code availability
Automatic Differentiation Model Builder (ADMB) version 12.3 was used for model fitting and simulation, MATLAB R2020b was used for visualisation and QGIS 3.20.3 was used for study site map. For queries about the model code please email J.G.D.R.

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

## Acknowledgements
We thank the Association of Marine Park Tourism Operators, Eye on the Reef Program, Great Barrier Reef Marine Park Authority, Dr Cameron Fletcher and Dr David Westcott for their roles in the provision of data that we fitted our model to. Figure 1 was designed by Dr Stacey McCormack (Visual Knowledge). We also extend thanks to Dr Laura Blamey, Dr Scott Condie and Prof. Anthony Richardson who reviewed and made constructive comments on the manuscript which led to its improvement. The present study forms part of the PhD work of J.G.D.R., which was funded through scholarships provided by the University of Queensland and the CSIRO.

## Author contributions
J.G.D.R. and E.E.P. formulated the present study, conducted analysis and revised the manuscript. J.G.D.R. developed and ran the model and wrote the first draft of the manuscript.

## Competing interests
The authors declare no competing interests.
