## [Peer Review File · Nature Communications]

Reviewers' comments:

Reviewer #1 (Remarks to the Author):

Review of Rogers & Plaganyi "Culling corallivore improves short-term coral recovery under bleaching scenarios"

General comments

The study by Rogers & Plaganyi sets out to evaluate the response in coral reef cover under several Crown-of-Thorns -Starfish culling strategies. The research is predicated on a strong cause and effect relationship between starfish culling, measured as "Catch-Per-Unit-Effort", and a positive response in coral cover. The key contextual background for the study is stated as: "...whether corallivore (starfish) management does or does not improve coral cover, which is necessary for large-scale operationalisation, remains equivocal". I was therefore expecting the study to provide an unequivocal result as to whether corallivore management could achieve an improvement in coral cover. However, I did not see evidence presented to demonstrate this. Instead, the relationship between culling of starfish and improved coral cover, which formed the basis of all subsequent modelling, was correlative only. Given this assumed relationship and an absence of prior critical demonstration of the effectiveness of culling on improved coral cover in a controlled experimental sense (e.g., Before-After-Control-Impact study), it is therefore impossible to critically conclude that culling improves short-term coral recovery under various bleaching scenarios. An alternative reason that declining CPUE could appear linked to increasing coral cover is that the starfish undergoes boom and bust cycles and so if a bust was coincident with a period of coral recovery post-bleaching, it may appear that starfish decline was driving coral recovery, but this assumption would be incorrect.

Unfortunately, as this modelling study is currently pitched, it appears incapable of moving beyond the equivocal nature of its key contextual statement. Given this the pitch and conclusions of the study needs to be sufficiently toned-down to categorically acknowledge the shortcomings of the data modelled. Otherwise the manuscript is very well written, well analysed and well supported by figures and tables.

Specific comments

- CPUE is a relative index of starfish abundance. Moreover, it is an index of the number of starfish killed and not the number of starfish that remain on the reef. Thus, linking decline in CPUE and improvement in coral cover appears outwardly challenging and not empirically defined.
- CPUE also varies in response to catchability, or in this case detectability, of the starfish by culling divers. For CoTS, detectability of individual starfish (and thus CPUE) may decrease at high coral cover, particularly high cover of the preferred coral prey of *Acropora* sp plate and branched coral types, which provide refuge whereby starfish can remain hidden within yet repeatedly emerge to form local feeding scars on the coral.
- Change in the frequency of coral scarring would be a way of examining relationships between increasing CPUE and reduced coral cover. Hopefully such data is available.
- Impacts of predators on prey populations can also be lagged in time, which would be expected in the

situation where CPUE decline may ultimately indicate declining starfish populations and therefore ultimately lead to positive responses in coral cover over subsequent years. In contrast, CPUE would be expected to decline immediately if increasing coral cover (due to other means such as post bleaching recovery) was to obscure detectability and thus CPUE of starfish. Conversely, the presence of bleached coral may result in higher CPUE as starfish may aggregate on few remaining preferred coral prey which could be more easily detected by starfish culling divers.

- Regarding the statement “Removal of invertebrate corallivores has been demonstrated to support coral thermal resistance and recovery following bleaching events¹⁴” – this implies that the effectiveness of starfish culling on coral recovery is unequivocal (thereby seemingly addressing the prior statement regarding the unequivocal nature of the effect of starfish culling), however reference no. 14 does not pertain to the starfish, but rather a gastropod. Furthermore, this study is conducted at a very small scale and the scalability of corallivorous gastropod control is unclear relative to the scale modelled by Roger & Plaganyi.
- The statement “Here we show intensive management (culling) of an invertebrate corallivore, *Acanthaster cf. solaris*, improved coral cover at sub-reef spatial scales” is not supported by the results. Alternatively, this statement and the entire study could be rephrased to indicate that intensive management ‘could’ improve coral cover at sub-reef spatial scales. As a modelling only study, I struggle to see how it can say more than that.

Reviewer #2 (Remarks to the Author):

General comments

From a global point of view, this is an original and rather innovative study based on an impressive modeling work. The different working hypotheses on the ecology of *acanthaster* populations, coral biology and climatic hazards (cyclones, thermal stress) are consistent with recent data from the literature, and are presented in a particularly clear and detailed manner. I would like to congratulate the authors for the meticulous work of modeling the different ecological processes and their interactions, which significantly enrich the scope of the model and its results. The presentation of the hypotheses in the Materials & Methods section is particularly rich in information, and allows a better understanding of the extent of the bibliographic synthesis work provided. Overall, I have very few comments/suggestions.

Remarks & suggestions

- I would suggest keeping Table S1 (Summary of intervention strategies) in the main document, and not as supplementary material. It allows for better interpretation of the results without requiring constant back and forth with the text.
- It would be useful to provide a map of the GBR area from which the main dataset (reefs & management sites) was derived, in order to better understand the spatial scales considered for the modeling of the different processes (in particular biological processes related to COTS populations/corals vs. climatic processes, e.g. cyclones).
- The number of supplementary figures is particularly high (86 figures!). Although these figures facilitate

a finer interpretation of the results, it is questionable whether a detailed reading of the coral cover and CPUE at site level is really essential. Could we consider, for example, grouping these results using a typology of sites, in the same way as for the presentation of the results of sites 2 and 7, selected "for their near median response to thermal stress 81 perturbation scenarios (Fig. 2-3)"?

- This work is based on a highly spatially contextualized dataset. In your opinion, to what extent would the results obtained be transposable to other geographical areas within the range of the species *A. solaris*?

Reviewer #3 (Remarks to the Author):

Overall, I think this is an interesting and timely study that advances our ability to understand, predict, and optimize coral reef management strategies. The manuscript is generally well-written, but does need to be revised in certain areas to improve readability. It would be great to be able to ground truth how well these modeling efforts reflect real-world coral recovery trajectories, but I understand that the amount of time that has elapsed is a potential constraint. Is there a reason why more recent data (up to ~2020) were not included (COVID constraints?)?

That said, I think once the authors address comments from myself and others (I will be interested to see the perspective of other reviewers) that the manuscript could be suitable for publication in *Nature Communications*.

Please see my detailed comments in the attached PDF.

Responses to Reviewer comments

Reviewer #1:

The study by Rogers & Plaganyi sets out to evaluate the response in coral reef cover under several Crown-of-Thorns -Starfish culling strategies. The research is predicated on a strong cause and effect relationship between starfish culling, measured as “Catch-Per-Unit-Effort”, and a positive response in coral cover. The key contextual background for the study is stated as: “...whether corallivore (starfish) management does or does not improve coral cover, which is necessary for large-scale operationalisation, remains equivocal”. I was therefore expecting the study to provide an unequivocal result as to whether corallivore management could achieve an improvement in coral cover. However, I did not see evidence presented to demonstrate this. Instead, the relationship between culling of starfish and improved coral cover, which formed the basis of all subsequent modelling, was correlative only. Given this assumed relationship and an absence of prior critical demonstration of the effectiveness of culling on improved coral cover in a controlled experimental sense (e.g., Before-After-Control-Impact study), it is therefore impossible to critically conclude that culling improves short-term coral recovery under various bleaching scenarios.

An alternative reason that declining CPUE could appear linked to increasing coral cover is that the starfish undergoes boom and bust cycles and so if a bust was coincident with a period of coral recovery post-bleaching, it may appear that starfish decline was driving coral recovery, but this assumption would be incorrect.

We appreciate the Reviewer’s careful consideration of these two points but would like an opportunity to rebut these criticisms as detailed below, and have also revised our manuscript accordingly to strengthen the basis for our findings.

We expand further below on the point around use of CPUE, but would also like to challenge the Reviewer’s statement that “ the relationship between culling of starfish and improved coral cover, which formed the basis of all subsequent modelling, was correlative only” – this is not correct although we appreciate that this misunderstanding may be because due to space limitations this aspect of the modelling may not have been explained in sufficient detail and we have addressed this in our revision (see lines 732-740).

“Simultaneously fitting CoTS and coral dynamics at concurrent locations was useful here as it allowed for coral cover trajectories to help inform local CoTS abundance (sensu CoTS feeding vs. coral trajectories (Keesing and Lucas, 1992, Plagányi et al., 2020) and local site fidelity (Ling et al., 2020)). Our model also used Long Term Monitoring Program (LTMP) data (based on manta tows and provided by the Australian Institute of Marine Science) which provides an independent index of relative abundance of CoTS. This was such that our model here was developed and parametrised based on an earlier version (Plagányi et al., 2014, Morello et al., 2014) which did not use CPUE information but were fitted to the LTMP data on CoTS relative abundance, as well as corresponding coral cover, to estimate a number of CoTS-coral interaction parameters used in the present model (Table S3).”

We are therefore confident as to the parametrisation of how changes in CoTS influence corals. Below we explain the basis for linking changes in CPUE to changes in CoTS (and hence to corals) and hence why our results have a solid basis and are not simply correlative.

Second, whilst we agree with the Reviewer that *Acanthaster* can, and do, undergo boom-bust dynamics, this generally occurs over a period of years (build-up) or are the consequence of a lagged recruitment event (Pratchett et al., 2014). Our study takes account of that fact that CoTS take almost

2 years from settlement before they are large enough to be detected (and targeted by a control program) and hence the boom-bust cycle cannot observably operate faster than a few years (e.g. ~15 year cycle on the GBR, Condie et al., 2018) whereas our study is analysing impacts of changes in CoTS on corals on much shorter time-scales. Similarly, coral recovery operates on the timescale of years (Morello et al., 2014, Condie et al., 2018) and we also explicitly accounted for low coral cover linked to increased CoTS mortality (i.e. a potential negative CoTS trend when coral abundance is low). Relatively speaking, the timescale of our data (multiple data points per year) is sufficiently fast that this statement is contextually false (i.e. there is sufficient resolution to capture trends) and our model was structurally capable of capturing increasing coral cover and decreasing/suppressed CoTS – indeed, this last point underpins one of our findings (limited culling impact if coral is low).

Unfortunately, as this modelling study is currently pitched, it appears incapable of moving beyond the equivocal nature of its key contextual statement. Given this the pitch and conclusions of the study needs to be sufficiently toned-down to categorically acknowledge the shortcomings of the data modelled. Otherwise the manuscript is very well written, well analysed and well supported by figures and tables.

Specific comments

CPUE is a relative index of starfish abundance. Moreover, it is an index of the number of starfish killed and not the number of starfish that remain on the reef. Thus, linking decline in CPUE and improvement in coral cover appears outwardly challenging and not empirically defined.

We agree with the Reviewer that CPUE is a relative index of starfish abundance but not that it is only an index of the number of starfish killed (the latter is in fact given by the catch). CPUE is widely used globally as an index of the relative abundance of fishery species (noting that associated concerns such as evidence for hyperstability are also frequently discussed) and our study uses a similar approach that has been carefully considered as detailed below. As with fisheries models, the total catch is the absolute number of removals (here starfish killed) but the catch-per-unit-effort is a rate which varies as a function of the underlying population abundance – hence if the population is more abundant the rate will be higher vs a lower catch rate when the population is reduced. “Our model doesn’t only represent the removals, but has an underlying (validated) population dynamics representation which dynamically updates the estimated changes in the population abundance due to the net effect of growth, recruitment, mortality and removals” (added text to lines 71-73:). It is these changes in the relative CoTS population size that we fit to the CPUE observations, and hence we use the information to estimate the number of starfish that remain on the reef (i.e. it is not correct that CPUE is only an index of the no. killed). As explained above, we then use a previously validated relationship between CoTS relative abundance and coral cover to quantify the impact of the remaining CoTS on the coral.

We have used multiple lines of evidence to validate the relationship used between CPUE and relative CoTS abundance as follows:

1. “A recent study by MacNeil et al. (2016) inter-calibrated multiple data sources (including mark-recapture data) and found the AMPTO CPUE observations (the same ones we use in our study) to be informative of reef-scale average CoTS densities” (added to lines 419-422).
2. Our analyses were also informed by an earlier study (described in Plagányi et al., 2020, Babcock et al., 2014) based on an empirically derived relationship using data collected by Fisk and Power (1999). The data comprised simultaneous catch per unit effort (COTS removed·h⁻¹) and COTS density estimates (COTS·ha⁻¹) from two small reefs around Lizard Island between October 1995 and August 1996 and each reef was visited weekly for the first ten weeks and then every 2 weeks for the following 20 weeks to quantify the relationship

between CPUE and CoTS density.

“We incorporated a formally fitted hyperstability relationship to represent management intervention (Plagányi et al., 2020). This was based on an empirically derived relationship using data collected by Fisk and Power (1999) which comprised simultaneous CPUE (CoTS.h⁻¹) and CoTS density estimates (CoTS.ha⁻¹) over a period of 20 weeks during 1995-1996.” (added to lines 416-419);

3. Both these studies (MacNeil et al. 2016; Plagányi et al. 2020 and see also Babcock et al. 2014) found the relationship between CPUE and known CoTS density to be hyper-stable and we use this empirically validated hyperstable formulation in our model.

“Both MacNeil et al. (2016) and Plagányi et al. (2020) found the relationship between CPUE and known CoTS density to be hyperstable; we used the empirically validated relationship of Plagányi et al. (2020) as it also comprised smaller CoTS compared to MacNeil et al. (2016).” (added to lines 422-424);

4. Unlike typical surveys, the culling program we describe controlled sites until no more starfish were detectable (Westcott et al., 2020). This often involved repeated site visits to ensure a high likelihood of removing individuals that were previously cryptic. This was enhanced by divers helping to identify potentially cryptic starfish through their feeding scars (Great Barrier Reef Marine Park Authority, 2017). Starfish likely to be located nearby their feeding scars (Ling et al., 2020, Great Barrier Reef Marine Park Authority, 2017, Keesing and Lucas, 1992).

“Detection rates (as perceived on SCUBA) are predominantly a function of CoTS size and population abundance (MacNeil et al., 2016, Plagányi et al., 2020). Recent studies have found little evidence for coral cover (MacNeil et al., 2016) or reef structural complexity (Kayal et al., 2017) influencing detection rates (using SCUBA). This is likely due to divers being able to search the reef matrix for hidden CoTS unlike manta tow based approaches (Westcott et al., 2021a). Feeding scars (which are generally a sign of proximate CoTS (Keesing and Lucas, 1992, Ling et al., 2020)) are also used by culling program divers to help locate individuals hidden in the reef matrix (Great Barrier Reef Marine Park Authority, 2017). Additionally, repeated site visits (as is the case for the sites we consider here) limit the number of CoTS that go undetected at a location (Westcott et al., 2020). The detection rates described above (i.e. 19%, 82%, and 82%), were therefore applied to the CoTS population within a management site independent of coral cover or structural complexity. This was such that if an individual was observable (given size-dependent detection and population abundance biases; described below under “CoTS management intervention”) then it was potentially controllable.” (added to lines 330-340).

We also deliberately used sites that had high visitation rates. We are therefore confident that declining and low CPUE values fairly accurately reflect low CoTS abundance (detectable CoTS);

5. “We used empirically derived estimates of size-age detectability to model the number of starfish within a given population that would be available to control on a particular day. This accounted for the fact that some starfish would still be undetectable and remain on the reef. To this extent, *Acanthaster spp.* are voracious coral predators and it is unlikely that a large number of undetected individuals would not alter coral cover trajectories (Keesing and Lucas, 1992)” (added to lines 324-328).

6. We fitted our model – which coupled starfish and coral dynamics – simultaneously to both catch rate and coral cover data at concurrent locations. In other words, if individuals were missed by a control team, the simultaneous fitting to coral cover data helps to characterise the size of said *A. c.f. solaris* population.

“Simultaneously fitting CoTS and coral dynamics at concurrent locations was useful here as it allowed for coral cover trajectories to help inform local CoTS abundance (sensu CoTS feeding

vs. coral trajectories (Keesing and Lucas, 1992, Plagányi et al., 2020) and local site fidelity (Ling et al., 2020)).” (added to lines 732-734).

7. Based on the literature, CoTS densities are extensively and comprehensively documented as a cause of widespread coral cover decline and hence there is a solid theoretical underpinning of our model.

We are therefore confident that we have a well substantiated basis to use CPUE as a reasonable proxy for starfish density in our study.

CPUE also varies in response to catchability, or in this case detectability, of the starfish by culling divers. For CoTS, detectability of individual starfish (and thus CPUE) may decrease at high coral cover, particularly high cover of the preferred coral prey of *Acropora* sp plate and branched coral types, which provide refuge whereby starfish can remain hidden within yet repeatedly emerge to form local feeding scars on the coral.

We agree with the Reviewer that CPUE does vary with detectability (which we explicitly modelled) but we disagree that there is sufficient evidence to support the Reviewer’s comments about some of the listed factors. These points are refuted by recent studies such as MacNeil et al. (2016) who found little evidence for an effect of hard coral cover on that detectability. Detectability (using SCUBA) has been found to not vary with reef matrix complexity (Kayal et al., 2017). The divers of the control program use feeding scars to help locate starfish (Great Barrier Reef Marine Park Authority, 2017) and starfish are typically located nearby their feeding scars (Ling et al., 2020, Keesing and Lucas, 1992, Great Barrier Reef Marine Park Authority, 2017). Repeated control (as in the case of our sites) also limits the number of cryptic CoTS that remain (Westcott et al., 2020). Detectability is predominantly influenced by starfish size (Plagányi et al., 2020, MacNeil et al., 2016) which we modelled (as detailed under point 4 above, referencing lines 330-340).

Change in the frequency of coral scarring would be a way of examining relationships between increasing CPUE and reduced coral cover. Hopefully such data is available.

The Reviewer raises an interesting point – this is actually a priority for future CoTS monitoring and surveillance on the Great Barrier Reef (part of the development and use of towed underwater vehicles, (Westcott et al., 2021a)). However, contemporary data collected on scarring is insufficient to infer starfish density and (in its current surveillance and monitoring use with manta tows) is an indicator to trigger culling (one or more scars on a manta tow) at a location where there may otherwise be cryptic individuals (Westcott et al., 2021a, Fletcher et al., 2020). We do note that CoTS feeding scars are also used to help locate cryptic individuals during culling dives (Great Barrier Reef Marine Park Authority, 2017). To address concerns about the role of feeding scars, we have added to lines 333-335:

“Feeding scars (which are generally a sign of proximate CoTS (Keesing and Lucas, 1992, Ling et al., 2020)) are also used by culling program divers to help locate individuals hidden in the reef matrix (Great Barrier Reef Marine Park Authority, 2017).”

Impacts of predators on prey populations can also be lagged in time, which would be expected in the situation where CPUE decline may ultimately indicate declining starfish populations and therefore ultimately lead to positive responses in coral cover over subsequent years. In contrast, CPUE would be expected to decline immediately if increasing coral cover (due to other means such as post

bleaching recovery) was to obscure detectability and thus CPUE of starfish. Conversely, the presence of bleached coral may result in higher CPUE as starfish may aggregate on few remaining preferred coral prey which could be more easily detected by starfish culling divers.

Detectability has been found to not vary with coral cover (MacNeil et al., 2016) and both coral cover data and CPUE data are used to inform starfish density in our model (incorporated as above in responses). We acknowledged in our discussion that we did not capture movement within our model (by which increased CPUE would theoretically occur) and we discuss results in relation to this caveat (lines 180-202). It is also of note, that whilst starfish are theoretically capable of potentially large displacements, the study in which this was found based this on a short-term escape response and it is unclear if such directional velocity can be sustained (Pratchett et al., 2017b). More recent work suggests that *A. c.f. solaris* exhibit limited net displacement (despite potential for ‘high velocity’ movement, Pratchett et al., 2017b) and homing like behaviour (site fidelity) under a range of coral cover levels (Ling et al., 2020). Where roaming behaviour (due to low prey availability) was detected, this was at a displacement of < 20 m in a day (Ling et al., 2020). Similarly, where aggregation of individuals following a bleaching event has been hypothesised, it was on a time scale of ~ 2 years (Haywood et al., 2019, Keesing et al., 2019).

Regarding the statement “Removal of invertebrate corallivores has been demonstrated to support coral thermal resistance and recovery following bleaching events¹⁴” – this implies that the effectiveness of starfish culling on coral recovery is unequivocal (thereby seemingly addressing the prior statement regarding the unequivocal nature of the effect of starfish culling), however reference no. 14 does not pertain to the starfish, but rather a gastropod. Furthermore, this study is conducted at a very small scale and the scalability of corallivorous gastropod control is unclear relative to the scale modelled by Roger & Plaganyi.

The context of this statement was intended to identify that removal of a corallivore (the gastropod) may support better coral cover outcomes under thermal stress and that it would be worthwhile investigating whether removing *A. c.f. solaris* does too. This sentence has been rephrased for better comprehension.

The sentence and its context now make this clear (lines 34-39): “Invertebrate coral predators, corallivores, are targeted by coral reef management authorities globally (Nakamura et al., 2016, Westcott et al., 2020, Chak et al., 2016, Shaver et al., 2018). Removal of an invertebrate corallivore (*Coralliophila abbreviata*) has been demonstrated to support coral thermal resistance and recovery following bleaching events (Shaver et al., 2018). The removal of another corallivore (*Acanthaster spp.*) are similarly posited as a viable means to increase coral reef resilience in other systems (Anthony et al., 2015, Wolff et al., 2018). In general, corallivore management is suggested to be a fundamental consideration in restoration initiatives ((e.g. Ladd and Shantz, 2020, Williams et al., 2014, Condie et al., 2021)).”.

We have also added to a “generality of findings” paragraph in the discussion that up-scaling corallivorous gastropod control is likely to be more difficult compared to CoTS given the relatively large scales at which CoTS control operates.

Specifically, we state (lines 254-257): “In relation to other species of invertebrate corallivore (e.g. gastropods), we highlight that their control is likely to also benefit from identifying local scale principles (Shaver et al., 2018, Williams et al., 2014), but may encounter greater difficulty in up-scaling culling efforts given the often relatively large scales of CoTS control efforts (e.g. (Westcott et

al., 2020, Yamaguchi, 1986, Westcott et al., 2021b)).”.

The statement “Here we show intensive management (culling) of an invertebrate corallivore, *Acanthaster cf. solaris*, improved coral cover at sub-reef spatial scales” is not supported by the results. Alternatively, this statement and the entire study could be rephrased to indicate that intensive management ‘could’ improve coral cover at sub-reef spatial scales. As a modelling only study, I struggle to see how it can say more than that.

As above, we feel our results are adequately supported by the combination of a modelling study that uses and is fitted to empirical data. Nonetheless, we have accounted for the Reviewer’s concerns by rephrasing this sentence as follows (lines 15-17): “Here we model intensive management (culling) of an invertebrate corallivore, *Acanthaster c.f. solaris*, and show that culling could improve coral cover at sub-reef spatial scales”. In doing this categorically acknowledges this is a modelling study and addresses that inferences and analysis were based on projections (i.e. ‘could’).

Reviewer #2 (Remarks to the Author):

General comments

From a global point of view, this is an original and rather innovative study based on an impressive modeling work. The different working hypotheses on the ecology of acanthaster populations, coral biology and climatic hazards (cyclones, thermal stress) are consistent with recent data from the literature, and are presented in a particularly clear and detailed manner. I would like to congratulate the authors for the meticulous work of modeling the different ecological processes and their interactions, which significantly enrich the scope of the model and its results. The presentation of the hypotheses in the Materials & Methods section is particularly rich in information, and allows a better understanding of the extent of the bibliographic synthesis work provided. Overall, I have very few comments/suggestions.

Response: We thank the Reviewer for their positive feedback. We have detailed how we have incorporated this below.

Remarks & suggestions

- I would suggest keeping Table S1 (Summary of intervention strategies) in the main document, and not as supplementary material. It allows for better interpretation of the results without requiring constant back and forth with the text.

We have moved Table S1 into the main document. This is now Table 2. Table references through the main and supplementary documents have been updated accordingly. The table has been renamed to "summary of modelled scenarios".

- It would be useful to provide a map of the GBR area from which the main dataset (reefs & management sites) was derived, in order to better understand the spatial scales considered for the modeling of the different processes (in particular biological processes related to COTS populations/corals vs. climatic processes, e.g. cyclones).

A map of the GBR has been added that delineates the reefs on which the sites were located (Fig. 2).

- The number of supplementary figures is particularly high (86 figures!). Although these figures facilitate a finer interpretation of the results, it is questionable whether a detailed reading of the coral cover and CPUE at site level is really essential. Could we consider, for example, grouping these results using a typology of sites, in the same way as for the presentation of the results of sites 2 and 7, selected "for their near median response to thermal stress 81 perturbation scenarios (Fig. 2-3)"?

We have redone the detailed site-specific plots in the supplementary figures as per main figures 2-3. This has substantially reduced the number of figures in the supplement from 86 down to 45.

- This work is based on a highly spatially contextualized dataset. In your opinion, to what extent would the results obtained be transposable to other geographical areas within the range of the species *A. solaris*?

We have added a paragraph to the discussion in response to the Reviewer's question (lines 239-257):

"While our work was based on a contextualised data set, we expect our results to have broader geographical relevance. It is however important to carefully consider generalisation. We note that most of our understanding comes from *A. cf. solaris* (Pratchett et al., 2017a) which is found through the Pacific and along the west Australian coast and that differences in outbreak severity and propensity may differ amongst *Acanthaster* spp. (Haszprunar et al., 2017). This requires further research to understand if, how, and what species differences influence population dynamics and

impacts (Pratchett et al., 2017a). We also mark that coral growth rates vary by coral species and by location which may influence modelled recovery rates (Morello et al., 2014, Pratchett et al., 2015). The degree to which recovery rates are impacted will primarily depend on coral growth rates compared to CoTS consumption rates (sensu Plagányi et al., 2020). Given our model structure (e.g. CoTS preference for faster growing corals, sequential depletion of coral taxa along preference gradients, and bleaching susceptibility of coral prey) is consistent with that of systems in which *A. cf. solaris* are problematic (Pratchett et al., 2014, Pratchett et al., 2017a), we expect robustness in the principle of CoTS management to reduce bleaching consequences; but, we also highlight the importance of carefully considering local conditions and the intrinsic capacity of corals to recover over a given timescale. Relative to intrinsic – local – coral recovery capacity, we expect our findings around CoTS control to be of general transposability within the range of *A. cf. solaris*. In relation to other species of invertebrate corallivore (e.g. gastropods), we highlight that their control is likely to benefit from identifying local scale principles (Shaver et al., 2018), but may encounter greater difficulty in up-scaling culling efforts given the often relatively large scales of CoTS control efforts (e.g. (Westcott et al., 2020, Yamaguchi, 1986, Westcott et al., 2021b)).”

Reviewer #3 (Remarks to the Author):

Overall, I think this is an interesting and timely study that advances our ability to understand, predict, and optimize coral reef management strategies. The manuscript is generally well-written, but does need to be revised in certain areas to improve readability. It would be great to be able to ground truth how well these modeling efforts reflect real-world coral recovery trajectories, but I understand that the amount of time that has elapsed is a potential constraint.

Response: We wish to thank the Reviewer for their detailed and helpful comments and feedback on our manuscript. We have addressed their feedback as detailed below.

Is there a reason why more recent data (up to ~2020) were not included (COVID constraints?)?

One the primary reasons that analyses were conducted up to 2018 (opposed to ~2020) was that the CoTS management program underwent a large structural change post-2018 (adoption of Integrated Pest Management principles, IPM). This saw differences in, for example, the physical size of management site, the number of sites considered in the program, and in the way CoTS control was conducted at reefs (Westcott et al., 2021b, Fletcher et al., 2020). Such differences could have feasibility impacted the reliability of extending the pre-IPM data series to include post-IPM data. Given this we opted for the longer time series (2013-2018) over the shorter series available from the post-IPM period at the time of analysis (i.e. 2019-2020). Aggregation to the reef scale would potentially address such concerns but also limit the rich insight into finer scale heterogeneity that we found in the present study. Theoretically, finer partitioning of reefs would allow use of our study to help to identify, more specifically, which areas of reefs may be in greater need of control efforts given the distribution of environmental conditions and resilience factors.

That said, I think once the authors address comments from myself and others (I will be interested to see the perspective of other reviewers) that the manuscript could be suitable for publication in Nature Communications.

Please see my detailed comments in the attached PDF.

Reviewer #3 PDF: Note that given previous feedback line number(s) have changed from the original submission so we have updated the line number(s) in brackets on our responses.

Line 1: Change to “Culling corallivores”

We have changed as suggested (line 1).

Line 28: It might be good to reword this to imply that you’re referring to refugial regions for corals. Given the short format of the manuscript, the message needs to be as clear to the reader as possible.

We have added that refugial regions are related to corals. Reworded as, “Managing portfolios of refugial regions for corals, such as Australia’s Great Barrier Reef 4, are likely to be” (lines 29-31).

Line 28: This should be “are likely.” Also I have skimmed the manuscript and there are many instances where the authors place emphasis on words, phrases, etc. If you are going to do this, it should be “—such as Australia’s Great Barrier Reef4—“Here, I think that you should just use commas: “Managing portfolios of refugial regions, such as Australia’s Great Barrier Reef4 , are likely...”

We have been through and reconsidered occurrences and replaced with commas, brackets, etc where appropriate.

Line 29: Change to “natural- or artificially-mediated”

We have changed as suggested (line 30).

Line 32: I would suggest rewording these last two sentences. the second part of the last sentence seems disconnected from the notion it is supposed to be connected to in the previous sentence.

We have reworded the last sentence to better link with its preceding sentence. As per Reviewer’s suggestion, this makes the paragraph more cohesive. Sentence has been changed to, “In these systems, local-scale interventions may reduce large-scale pressures that are less amenable to direct management, thereby providing more opportunity for coral reefs to adapt and acclimate 5,8-10.” (lines 31-33).

Line 39: At present, this doesn’t read well. You could change it to: “efforts to control the corallivorous crown-of-thorns starfish” or “efforts to control a corallivore—the crown-of-thorns starfish (*Acanthaster cf. solaris*)— on the Great Barrier Reef”

We have taken Reviewer’s advice and rephrased as per suggestion, specifically “... efforts to control the corallivorous crown-of-thorns starfish (CoTS, *Acanthaster c.f. solaris*), ...” (lines 41-43).

Lines 40-41: This creates further pauses in a sentence that doesn’t need them...in a manuscript that already has many pauses. I would say just write: “have been correlated with increased coral cover throughout a period that encompassed two major bleaching mortality events and elevated CoTS abundances.”

We have rephrased as per Reviewer’s suggestion (lines 43-44).

Line 41-42: Either use CoTS or *A. cf. solaris*, but stick with one. Switching back and forth between them is just added confusion for the reader.

We have gone through and replaced “*A. c.f. solaris*” with “CoTS” throughout the manuscript.

Line 42-43: These sentence needs to be clearer. What is meant by prey type? I know what you’re referring to, but make it easier for a reader who might not be familiar with CoTS.

We have clarified that “prey type” refers to coral species (line 45).

Line 44: Change to something like: “ecological risk to reef resilience to bleaching events”

We have rephrased more simply. “Therefore, they pose a serious ecological risk to the resilience of reefs to bleaching events by impeding recovery potential 25,26” has been changed to “Therefore, they pose a serious ecological risk to reef resilience to bleaching events by impeding recovery potential 25,26” (lines 47-48).

Line 48: I think it is probably better to just use the acronym CoTS here and throughout. Some are unfamiliar with the fact that *Acanthaster planci* has now been broken up into several different species.

We have gone through and replaced “starfish” with “CoTS” throughout the manuscript.

Lines 49-50: Change to: “during outbreaks has caused widespread loss of coral cover across many Indo-Pacific reefs29-31.”

This sentence has been rephrased as per Reviewer suggestion. “... during outbreaks causes widespread loss of coral cover across many coral reef systems in the Indo-Pacific” has been rephrased as “... during outbreaks has caused widespread loss of coral cover across many Indo-Pacific coral reefs” (line 53).

Line 66: I would change “starfish” to “CoTS” here and throughout the manuscript.

We have changed “starfish” to “CoTS” throughout manuscript.

Lines 116-117: Change to “However, the persistence of coral cover benefits is indicative”

We have changed in line with Reviewer suggestion (lines 124-125).

Line 125: I would change this to something like “Reduced rates of catch-per-unit effort were reduced due to”

This has been rephrased as per Reviewer suggestion. “Reduced rates of catch-per-unit-effort reduction were due to prey limitation. This is because prey limitation led to an increase in CoTS mortality rates.” (lines 133-134).

Line 127-128: Change to “Under the no thermal stress scenario (DHW = 0), coral cover...”

Line 127-130: This sentence isn’t clear and needs to be rewritten.

Lines 130-132: Again, the wording in some of these sentences is convoluted and could be streamlined.

Lines 131: I would change to “initially similar”

We have reworked the lines 127-131 in line with Reviewer’s suggestions.

Old lines:

Under the no thermal stress scenario (DHW=0) coral cover was higher than the alternative scenarios with the greatest management-derived benefit across alternative thermal stress scenarios in the initial period following the simulated bleaching event in year 2022 (Fig. 4). However, the derived coral cover benefit of intensive corallivore management under mild thermal stress perturbation (DHW=4) was similar and surpassed the benefit under the no thermal stress scenario (Fig. 4).

New lines (lines 135-139) :

Under the no thermal stress scenario ($DHW = 0$), coral cover was predictably higher than in the alternative scenarios where a bleaching event was simulated (e.g. Fig. 3-4). However, the coral cover benefit derived from intensive corallivore management under the mild thermal stress scenario ($DHW = 4$) was initially similar and then surpassed that of the no bleaching event scenario (Fig. 5).

Line 145: Either delete the “e.g.” or put it in parentheses with the citations.

We have put inside parentheses with the citations (line 152).

Lines 153-154: Should point #1 be separated into 2? 1) The capacity of the assemblages and 2) their thermal history?

We agree with Reviewer. Have split into two points (lines 160-161).

Lines 161: I would change this to “This suggests that, depending on prevailing withinsite conditions, management...”

We have changed “This suggests that dependent on prevailing within-site conditions, management” to “This suggests that, depending on prevailing within-site conditions, management” (lines 168-169).

Line 164: Change to “influence”

We have changed “also influence” to “influence” as suggested (line 171).

Line 165: The term “superposition” sounds odd here. I would use something like “interplay”

We have changed as suggested (line 172).

Lines 177-180: This is may not always be true, especially at the sub reef scales that are a focus of your study. As you mentioned earlier in the manuscript, Acanthaster can be highly mobile. In some cases, they may successfully recruit to certain locations and move into others to target viable coral populations (e.g. Clements and Hay 2014 PLOS One).

- We have added that our identification of sub-reef larval sinks is caveated by not considering movement/larval development. This is consistent with the existing message conveyed through the paragraph. This addition intends to clarify intent.

Old lines:

“Our model identifies sub-reef larval sinks as a key variable in management efficacy. This is because successful larval recruitment regulates/influences emergent corallivorous adult populations within management sites.”

New lines (lines 184-190):

“Our model identifies sub-reef larval sinks as a key variable in management efficacy, but this must be interpreted contextually with model caveats. This is because successful larval recruitment regulates and influences emergent corallivorous adult populations within management sites in our model. Whilst this is consistent with observed limited realised CoTS mobility (*sensu* displacement) and homing behavioural patterns (Pratchett et al., 2017b, Ling et al., 2020), the importance of sub-reef larval sinks could plausibly be influenced by adult movements among management sites (Pratchett et al., 2017b, Haywood et al., 2019, Ling et al., 2020, Clements and Hay, 2017) and/or variation in the development of juveniles (Deaker et al., 2020a, Deaker et al., 2020b).”

- We have added reference to Clements & Hay (2017) to assist contextualising potential consequences of movement – in referencing how movement influence sinks, and in how it could impact localised outbreaks.
- We have also added reference to Clements & Hay (2017) and Deaker et al. (2020) as examples in the considerations for CoTS population dynamics and local persistence for coral restoration initiatives. “... although such dynamics depend on corallivore populations and their local persistence (e.g. (Clements and Hay, 2017, Deaker et al., 2020a)).” (lines 230-231)

Line 180: “Up to 250-520m/day” is limited mobility? This statement seems to contradict your statement in the intro.

We have added in the intro that this is based on maximum velocity likely indicative of an escape response, but that under more typical circumstance movement (displacement opposed to velocity) is localised.

“... mobility (maximum escape response velocity indicative of up to 250-520m/day depending on substrate and prey availability though likely < 19m/day displacement) (Ling et al., 2020, Pratchett et al., 2017b)” (lines 45-46).

We have also added lines in the discussion (lines 187-190):

“Whilst this is consistent with observed limited realised starfish mobility (sensu displacement) and homing behavioural patterns 23,24, the importance of sub-reef larval sinks could plausibly be influenced by adult movements among management sites 23,24,26,51 and/or variation in the development of juveniles 52,53”

Lines 181-182: Ok, you did mention adult movement. In this case, I’m finding it hard to see how you can identify sub-reef larval sinks as the key variable. You don’t actually have data on larvae...just CPUE of adults.

Line 193: Ok, I see you address this towards the end of the paragraph.

Addressed as in above comments: Early in paragraph we now state our finding of sub-reef larval sinks as a key variable within the model and needs to be considered in the context of model caveats. As mentioned by Reviewer, we suggest in the manuscript that the consequence of these is likely a less pronounced role of sub-reef larval sinks and that better understanding of early life histories would help disentangle this. Specifically (lines 198-202): “Aggregative behaviours at the reef scale (i.e. across multiple sites) (Haywood et al., 2019) –such as during outbreaks– and/or prolonged juvenile phases which would theoretically allow ‘banks’ juveniles to build up, would likely reduce the role of sub-reef larval settlement patterns in the model. Nonetheless, our model results underscore the complex dynamics of sub-reef larval sinks (Wilmes et al., 2020) and adult movement behaviours (Pratchett et al., 2017b, Ling et al., 2020).”.

Lines 231-232: Revise this sentence.

We have deleted the line. Given previous two sentences it seemed redundant upon revision.

Lines 1114-1116: This needs to be rewritten.

These lines have been rewritten.

Old lines:

“Different levels of accumulated thermal stress are represented by differently coloured curves such that green (none, DHWs=0), blue (mild, DHW=4), orange (moderate, DHWs=7), and grey (high, DHWs=10).”

Rewritten as (lines 1161-1166):

“Different line colours correspond to different modelled levels of accumulated thermal stress experienced by corals. The green curve corresponded to no stress (DHWs=0), the blue curve corresponded to mild stress (DHW=4), the orange curve corresponded to moderate stress, (DHWs=7), and the grey curve corresponded to high accumulated thermal stress (DHWs=10). Figure panels vary in terms of modelled coral adaptive and acclimation capacity.”

Line 359: Make sure you go through the references and format them in a consistent manner. For example, there are multiple words capitalized in the title of #45. This is not the norm in the other references. You also have instances where journal titles are not capitalized properly (references #2, 5, 8, 9, 11, 12, 14, etc.)

We have been through and formatted titles and journal names through the reference list.

Literature cited

- ANTHONY, K. R., MARSHALL, P. A., ABDULLA, A., BEEDEN, R., BERGH, C., BLACK, R., EAKIN, C. M., GAME, E. T., GOOCH, M. & GRAHAM, N. A. 2015. Operationalizing resilience for adaptive coral reef management under global environmental change. *Global Change Biology*, 21, 48-61.
- BABCOCK, R., PLAGANYI, E., MORELLO, E., ROCHESTER, W., HOEY, J., PRATCHETT, M. & ANTHONY, K. 2014. What are the important thresholds and relationships to inform the management of COTS? Canberra, Australia: CSIRO.
- CHAK, S. T., DUMONT, C. P., ADZIS, K.-A. A. & YEWDALL, K. 2016. Effectiveness of the removal of coral-eating predator *Acanthaster planci* in Pulau Tioman Marine Park, Malaysia. *Journal of the Marine Biological Association of the United Kingdom*, 183-189.
- CLEMENTS, C. S. & HAY, M. E. 2017. Size matters: predator outbreaks threaten foundation species in small marine protected areas. *PLOS ONE*, 12, e0171569.
- CONDIE, S. A., ANTHONY, K. R. N., BABCOCK, R. C., BAIRD, M. E., BEEDEN, R., FLETCHER, C. S., GORTON, R., HARRISON, D., HOBDAV, A. J., PLAGÁNYI, É. E. & WESTCOTT, D. A. 2021. Large-scale interventions may delay decline of the Great Barrier Reef. *Royal Society Open Science*, 8, 201296.
- CONDIE, S. A., PLAGÁNYI, É. E., MORELLO, E. B., HOCK, K. & BEEDEN, R. 2018. Great Barrier Reef recovery through multiple interventions. *Conservation Biology*, 32, 1356-1367.
- DEAKER, D. J., AGÜERA, A., LIN, H.-A., LAWSON, C., BUDDEN, C., DWORJANYN, S. A., MOS, B. & BYRNE, M. 2020a. The hidden army: corallivorous crown-of-thorns seastars can spend years as herbivorous juveniles. *Biology Letters*, 16, 20190849.
- DEAKER, D. J., MOS, B., LIN, H.-A., LAWSON, C., BUDDEN, C., DWORJANYN, S. A. & BYRNE, M. 2020b. Diet flexibility and growth of the early herbivorous juvenile crown-of-thorns sea star, implications for its boom-bust population dynamics. *PLOS ONE*, 15, e0236142.
- FISK, D. A. & POWER, M. C. 1999. Development of cost-effective control strategies for crown-of-thorns starfish. CRC Reef Research Centre, Townsville.
- FLETCHER, C. S., BONIN, M. C. & WESTCOTT, D. A. 2020. An ecologically-based operational strategy for COTS control: integrated decision making from the site to the regional scale. Reef and Rainforest Research Centre Limited, Cairns
- GREAT BARRIER REEF MARINE PARK AUTHORITY 2017. *Crown-of-thorns starfish control guidelines*, Townsville, GBRMPA.
- HASZPRUNAR, G., VOGLER, C. & WÖRHEIDE, G. 2017. Persistent gaps of knowledge for naming and distinguishing multiple species of crown-of-thorns-seastar in the *Acanthaster planci* species complex. *Diversity*, 9, 22.
- HAYWOOD, M. D. E., THOMSON, D. P., BABCOCK, R. C., PILLANS, R. D., KEESING, J. K., MILLER, M., ROCHESTER, W. A., DONOVAN, A., EVANS, R. D., SHEDRAWI, G. & FIELD, S. N. 2019. Crown-of-thorns starfish impede the recovery potential of coral reefs following bleaching. *Marine Biology*, 166, 99.
- KAYAL, M., BOSSERELLE, P. & ADJEROUD, M. 2017. Bias associated with the detectability of the coral-eating pest crown-of-thorns seastar and implications for reef management. *Royal Society Open Science*, 4, 170396.
- KEESING, J. K. & LUCAS, J. S. 1992. Field measurement of feeding and movement rates of the crown-of-thorns starfish *Acanthaster planci* (L.). *Journal of experimental marine biology and ecology*, 156, 8994-91104.
- KEESING, J. K., THOMSON, D. P., HAYWOOD, M. D. & BABCOCK, R. C. 2019. Two time losers: selective feeding by crown-of-thorns starfish on corals most affected by successive coral-bleaching episodes on western Australian coral reefs. *Marine Biology*, 166, 72.
- LADD, M. C. & SHANTZ, A. A. 2020. Trophic interactions in coral reef restoration: a review. *Food Webs*, 24, e00149.

- LING, S., COWAN, Z.-L., BOADA, J., FLUKES, E. & PRATCHETT, M. 2020. Homing behaviour by destructive crown-of-thorns starfish is triggered by local availability of coral prey. *Proceedings of the Royal Society B*, 287, 20201341.
- MACNEIL, M. A., MELLIN, C., PRATCHETT, M. S., HOEY, J., ANTHONY, K. R., CHEAL, A. J., MILLER, I., SWEATMAN, H., COWAN, Z. L. & TAYLOR, S. 2016. Joint estimation of crown of thorns (*Acanthaster planci*) densities on the Great Barrier Reef. *PeerJ*, 4, e2310.
- MORELLO, E. B., PLAGÁNYI, É. E., BABCOCK, R. C., SWEATMAN, H., HILLARY, R. & PUNT, A. E. 2014. Model to manage and reduce crown-of-thorns starfish outbreaks. *Marine Ecology Progress Series*, 512, 167-183.
- NAKAMURA, M., HIGA, Y., KUMAGAI, N. & OKAJI, K. 2016. Using long-term removal data to manage a crown-of-thorns starfish population. *Diversity*, 8, 24.
- PLAGÁNYI, É. E., BABCOCK, R. C., ROGERS, J., BONIN, M. & MORELLO, E. B. 2020. Ecological analyses to inform management targets for the culling of crown-of-thorns starfish to prevent coral decline. *Coral Reefs*, 39, 1483-1499.
- PLAGÁNYI, É. E., ELLIS, N., BLAMEY, L. K., MORELLO, E. B., NORMAN-LOPEZ, A., ROBINSON, W., SPORCIC, M. & SWEATMAN, H. 2014. Ecosystem modelling provides clues to understanding ecological tipping points. *Marine Ecology Progress Series*, 512, 99-113.
- PRATCHETT, M. S., ANDERSON, K. D., HOOGENBOOM, M. O., WIDMAN, E., BAIRD, A. H., PANDOLFI, J. M., EDMUNDS, P. J. & LOUGH, J. M. 2015. Spatial, temporal and taxonomic variation in coral growth—implications for the structure and function of coral reef ecosystems. *Oceanography and Marine Biology: An Annual Review*. 1 ed.
- PRATCHETT, M. S., CABALLES, C. F., RIVERA-POSADA, J. A. & SWEATMAN, H. P. A. 2014. Limits to understanding and managing outbreaks of crown-of-thorns starfish (*ACANTHASTER* Spp.). In: ROGER N. HUGHES, D. J. H., AND I. PHILIP SMITH (ed.) *Oceanography and Marine Biology: An Annual Review* Taylor & Francis.
- PRATCHETT, M. S., CABALLES, C. F., WILMES, J. C., MATTHEWS, S., MELLIN, C., SWEATMAN, H., NADLER, L. E., BRODIE, J., THOMPSON, C. A. & HOEY, J. 2017a. Thirty years of research on crown-of-thorns starfish (1986–2016): scientific advances and emerging opportunities. *Diversity*, 9, 41.
- PRATCHETT, M. S., COWAN, Z.-L., NADLER, L. E., CABALLES, C. F., HOEY, A. S., MESSMER, V., FLETCHER, C. S., WESTCOTT, D. A. & LING, S. D. 2017b. Body size and substrate type modulate movement by the western Pacific crown-of-thorns starfish, *Acanthaster solaris*. *PLOS ONE*, 12, e0180805.
- SHAVER, E. C., BURKEPILE, D. E. & SILLIMAN, B. R. 2018. Local management actions can increase coral resilience to thermally-induced bleaching. *Nature Ecology & Evolution*, 2, 1075-1079.
- WESTCOTT, D. A., FLETCHER, C. S., GLADISH, D. W. & BABCOCK, R. C. 2021a. Monitoring and Surveillance for the Expanded Crown-of-Thorns Starfish Management Program. Report to the National Science Program. Cairns.
- WESTCOTT, D. A., FLETCHER, C. S., GLADISH, D. W., MACDONALD, S. & CONDIE, S. 2021b. Integrated Pest Management Crown-of-Thorns Starfish Control Program on the Great Barrier Reef: Current Performance and Future Potential. Report to the National Environmental Science Program. Cairns: Reef and Rainforest Research Centre Limited.
- WESTCOTT, D. A., FLETCHER, C. S., KROON, F. J., BABCOCK, R. C., PLAGÁNYI, E. E., PRATCHETT, M. S. & BONIN, M. C. 2020. Relative efficacy of three approaches to mitigate crown-of-thorns starfish outbreaks on Australia's Great Barrier Reef. *Scientific Reports*, 10, 1-12.
- WILLIAMS, D. E., MILLER, M. W., BRIGHT, A. J. & CAMERON, C. M. 2014. Removal of corallivorous snails as a proactive tool for the conservation of acroporid corals. *PeerJ*, 2, e680.
- WILMES, J. C., SCHULTZ, D. J., HOEY, A. S., MESSMER, V. & PRATCHETT, M. S. 2020. Habitat associations of settlement-stage crown-of-thorns starfish on Australia's Great Barrier Reef. *Coral Reefs*, 39, 1163-1174.

- WOLFF, N. H., MUMBY, P. J., DEVLIN, M. & ANTHONY, K. R. 2018. Vulnerability of the Great Barrier Reef to climate change and local pressures. *Global Change Biology*, 24, 1978-1991.
- YAMAGUCHI, M. 1986. *Acanthaster planci* infestations of reefs and coral assemblages in Japan: a retrospective analysis of control efforts. *Coral Reefs*, 5, 23-30.

REVIEWER COMMENTS

Reviewer #1 (Remarks to the Author):

The authors have performed a convincing rebuttal of my major concerns and they offer many general improvements to their manuscript. My original concerns, seemingly based on limited information available in the short format, have been allayed by the additional information that the authors have provided in their rebuttal and revised manuscript.

In summary, the authors satisfied my concerns regarding:

1. The use of CPUE and underlying assumptions.
2. Overreach of certain statements given the modelling only nature of the research, for which I appreciate the adjustment of tone in terms of the inferences, i.e. inclusion of the word 'could'.

Regarding point 2 above, I further encourage the authors to replace the two occurrences of "does" in the below sentence in the abstract with "can" and "cannot" respectively.

"However, whether corallivore management can or cannot improve coral cover, which is necessary for large-scale operationalisation, remains equivocal."

The reason being is that the model simulations enable the potential capacity of different management actions to be assessed, as opposed to the explicit phrasing of "does or does not" which requires unequivocal real-world proof that this is the case, which would require an actual large-scale and long-term manipulative experiment with/ without culling interventions.

Overall, the authors have adequately addressed my concerns and I acknowledge the editor for giving the authors the opportunity to do so.

Reviewer #4 (Remarks to the Author):

Dear Editor,

I have read the comments of Reviewer 1 as well as the comprehensive rebuttal of the authors. I must say that overall I think that they convincingly and directly addressed the reviewer's criticisms, which at times seemed more rhetorical than data-based---and the authors appropriately rebutted those claims using up-do-date literature.

The question I guess is whether this constitutes Nat Comm material. I think it might be. The modelling is very advanced and carefully delineated, and the authors now have toned down the conclusions as

pointed out by the Reviewer, properly reflecting that the conclusions of their study arose from a modelling exercise. The geographical scope is small, however the time series is quite extensive, which adds great value to their study.

I also believe that comments made by the Reviewer such as "I was therefore expecting the study to provide an unequivocal result" are harmful to science, and somehow add misplaced judgment because such a call should be made solely by the editor in light of (what should have been) an unbiased review. If anything, the authors are very careful, including in their abstract, to recognise when the evidence is strong, and it is weak. So I commend them for doing so.

The only thing I might suggest is for the authors to carefully consider, within the data that is available to them, whether they could run a BACI-type analysis. They would have to make some assumptions about what would constitute a good control reef, but I think this comment made by Reviewer 1 is important and should not be dismissed (I think it was). Here, an "impact site" would be represented by reefs affected by outbreaks where culling occurred, whereas a "control site" could be a nearby reefs suffering from an outbreak but where culling did not occur. I think LTMP data is comprehensive enough (spatially and temporally) for them to extract such cases for a sensitivity analysis. However, I would still disagree with the reviewer that a BACI could "critically conclude that culling improves short-term coral recovery under various bleaching scenarios". Although it is still a great analytical methodology for environmental risk assessment, the removal of CoTS (and a potential increase in coral cover post culling) could still be due to a third, perhaps additional, unmeasured variable (e.g. fish abundance) In that sense, authors could also use some covariates (e.g. environmental SST, fish abundance) to control for some of those potential effects.

Reviewer #1 (Remarks to the Author):

The authors have performed a convincing rebuttal of my major concerns and they offer many general improvements to their manuscript. My original concerns, seemingly based on limited information available in the short format, have been allayed by the additional information that the authors have provided in their rebuttal and revised manuscript.

In summary, the authors satisfied my concerns regarding:

1. The use of CPUE and underlying assumptions.
2. Overreach of certain statements given the modelling only nature of the research, for which I appreciate the adjustment of tone in terms of the inferences, i.e. inclusion of the word 'could'.

Regarding point 2 above, I further encourage the authors to replace the two occurrences of "does" in the below sentence in the abstract with "can" and "cannot" respectively.

"However, whether corallivore management can or cannot improve coral cover, which is necessary for large-scale operationalisation, remains equivocal."

The reason being is that the model simulations enable the potential capacity of different management actions to be assessed, as opposed to the explicit phrasing of "does or does not" which requires unequivocal real-world proof that this is the case, which would require an actual large-scale and long-term manipulative experiment with/ without culling interventions.

Overall, the authors have adequately addressed my concerns and I acknowledge the editor for giving the authors the opportunity to do so.

Reviewer #1 response:

We thank the Reviewer for reviewing our manuscript again and for their additional comments. We have adjusted the phrasing around point 2 as suggested.

Changes:

- Lines (13-14) now read:
"Whether corallivore management can improve coral cover –necessary for large-scale operationalisation– remains equivocal."

Reviewer #4 (Remarks to the Author):

Dear Editor,

I have read the comments of Reviewer 1 as well as the comprehensive rebuttal of the authors. I must say that overall I think that they convincingly and directly addressed the reviewer's criticisms, which at times seemed more rhetorical than data-based---and the authors appropriately rebutted those claims using up-to-date literature.

The question I guess is whether this constitutes Nat Comm material. I think it might be. The modelling is very advanced and carefully delineated, and the authors now have toned down the conclusions as pointed out by the Reviewer, properly reflecting that the conclusions of their study arose from a modelling exercise. The geographical scope is small, however the time series is quite extensive, which adds great value to their study.

I also believe that comments made by the Reviewer such as "I was therefore expecting the study to provide an unequivocal result" are harmful to science, and somehow add misplaced judgment because such a call should be made solely by the editor in light of (what should have been) an unbiased review. If anything, the authors are very careful, including in their abstract, to recognise when the evidence is strong, and it is weak. So I commend them for doing so.

The only thing I might suggest is for the authors to carefully consider, within the data that is available to them, whether they could run a BACI-type analysis. They would have to make some assumptions about what would constitute a good control reef, but I think this comment made by Reviewer 1 is important and should not be dismissed (I think it was). Here, an "impact site" would be represented by reefs affected by outbreaks where culling occurred, whereas a "control site" could be a nearby reefs suffering from an outbreak but where culling did not occur. I think LTMP data is comprehensive enough (spatially and temporally) for them to extract such cases for a sensitivity analysis. However, I would still disagree with the reviewer that a BACI could "critically conclude that culling improves short-term coral recovery under various bleaching scenarios". Although it is still a great analytical methodology for environmental risk assessment, the removal of CoTS (and a potential increase in coral cover post culling) could still be due to a third, perhaps additional, unmeasured variable (e.g. fish abundance) In that sense, authors could also use some covariates (e.g. environmental SST, fish abundance) to control for some of those potential effects.

** PLEASE FEEL FREE TO PASS THIS ON TO THE AUTHORS **

Reviewer #4 response:

We thank the Reviewer for their review and feedback on our manuscript.

Whilst we feel that the Reviewer makes a good point, we feel it would be difficult to properly address here and would likely be better suited to an additional empirical study rather than inclusion within the current modelling study.

To this extent, we have addressed some of the feedback on conducting a BACI-type analysis through including a sensitivity based on model outputs over the data period of 2013 to 2018. Specifically, we have used our model to compare the fitted model trajectories with simulations where manual control is removed. That is, the fitted trajectory is compared to a trajectory where we remove CoTS control. This allowed us to model how the amount of control effort expended at a site, the length of time between voyages to a site, variable CoTS recruitment, and various effective bleaching-induced mortality levels concurrently influenced the ability of CoTS control to improve coral trajectories. The outcomes of this exercise were comparable results (but of a lower median) to the projections (2019 to 2029) upon which we base our findings in the manuscript. This addresses some of the feedback around inclusion of a BACI-type analysis but also adds to understanding of the retrospective efficacy of CoTS control on coral cover trajectories. Further to this, in our discussion we have added an explicit recommendation for a BACI (or similar) type study to be conducted for CoTS control on the GBR to contribute additional insight.

Changes to manuscript in response:

- Section in methods (lines 809-819):

“Sensitivity to CoTS control

To test whether our projected scenarios were consistent with the period over which data were collected, we conducted a model-based before and after comparison to the impact of control. Specifically, we used the fitted trajectory for sites, including both the coral data and CoTS control data (voyages and time spent), and compared this to the model-suggested coral trajectories if CoTS control had not taken place. These were modelled over the fitted period (2013 to 2018) and, unlike the projected scenarios (2019 to 2029), were variable in terms of the timing of control (amount of time between visits was variable), the amount of time spent at sites (not a consistent number of dive minutes per visit), CoTS dynamics (recruitment was fitted and hence different annually and between reefs), and in the level of thermal stress they experienced (different sites experienced different effective levels and some sites experience back-to-back events).”

- Section in results (147-161):

“Sensitivity for model outputs over 2013 to 2018

Contrasting fitted model trajectories that included CoTS control, with simulations that did not (i.e. simulated Management Sites as if control had not taken place), suggest that there was a median improvement of ~1% in coral cover (mean was higher at ~2.1 %) over the five years of 2013-2018 (Fig. 7a). The bleaching event of 2016 and 2017 substantially reduced the signal of CoTS control (in terms of coral cover) in modelled trajectories (Fig. 7a-b). Prior to

the bleaching events, manual control is suggested by the model to have improved median coral cover over years 2013 to 2015 by $\sim 1.9\%$ (mean was $\sim 3.3\%$) but this was quickly reduced to a median of $\sim 0.7\%$ (mean was $\sim 1.9\%$). Predominantly due to the bleaching events, we found that the no control scenario indicated that median coral cover for the group of Management Sites in 2018 (relative to 2013) would likely have declined by $\sim 23\%$ without CoTS control but by a lesser amount of a median of $\sim 18\%$ with control (Fig. 7b-c). In terms of relative change at individual Management Sites over 2013 to 2018, the median of the differences was $\sim 3\%$ (Fig. 7d). That is, considering the bleaching events in 2016 and 2017, CoTS control is suggested by the MICE to have increased coral cover by 3% relative to no-control over the timeframe.”

- New figure – fig. 7 displaying results of sensitivity:

Fig. 7 Sensitivity for model outputs over the data period of 2013 to 2018. Figure is based on model outputs over 2013 to 2018 (June used as observed data ranged from mid-2013 to mid-2018 for most sites) which used undertaken control efforts by the CoTS Management Program as input (e.g. site visits and dive time) to inform potential coral trajectories in its absence. Sensitivity considered no adaptive and acclimation capacity ($A=0$). Abbreviation ‘C’ denotes ‘Control’ (CoTS culling) and ‘NC’ denotes ‘No Control’. (a) Summary of differences in coral cover between management scenarios of no manual control and inputted observed manual control over years 2013 – 2018. Difference represents the (absolute) difference in coral cover expressed as a percentage as opposed to a proportional difference. Both mean and median differences are plotted alongside the management-induced difference at each Management Site considered. (b) Box and whisker plot indicating median coral cover model output over years 2013 – 2018 with and without control. (c) Box and whisker plot indicating the model-suggested relative change in coral cover for 2018 relative to 2013 with and without control, positive values indicate an increase, negative values indicate a decrease, and 0 indicates no change. (d) Box and whisker plot indicating model-suggested relative change over 2013 – 2018 due to operation on the CoTS control program.

- Additional text in discussion (250-268):

“As a sensitivity we compared projected scenarios to the modelled trajectories over the data fitted period with and without CoTS control. We found the benefit of control persisted under the sensitivity although it was reduced over time (median improvement in coral cover of ~1 % over five years). The MICE suggested this to be due to the impacts of the 2016 and 2017 bleaching events. These events reduced the median of control-mediated improved coral cover by ~1.2 %. Overall, the MICE over 2013 to 2018 was consistent with the findings of our projection scenarios for 2019 to 2029 (improved up to a median of ~2 %, high levels of thermal stress reduced detectable management signal).

Ideally, a before-after-control-impact (BACI; or similar) study (e.g. ⁶⁶) would contribute to understanding of CoTS control on coral trajectories. An empirical BACI analysis should be a priority for future CoTS GBR control work and could potentially be achieved by leveraging and combining multiple sources of data (e.g. ⁶⁷) such as from the AIMS LTMP data and recent Integrated Pest Management CoTS control program. Based on our modelling study, it is strongly recommended that care is taken in how to define ‘control’ sites and how well they are likely to track an impact site where CoTS management takes place. The present modelling suggests that the signal of CoTS control is heterogenous depending on local factors and sensitive to perturbation (bleaching). The 2016 and 2017 bleaching events have also been suggested by Westcott, et al. ¹² to have reduced the full benefit of CoTS control in terms of coral cover. A BACI analysis would provide additional insight into how CoTS control influences coral cover trajectories.”

Other changes (formatting, data/code availability etc.):

- Results subheading “Management effectiveness varied across management sites and thermal stress levels” shortened to “Management effectiveness varied” to comply with 60 character limit.
- Subheadings removed from discussion section.
- In acknowledgements sections have added funding sources for PhD scholarships – added to end of sentence “... which was funded through scholarships provided by the University of Queensland and the CSIRO”.
- Acronyms defined in figure captions.
- Updated references to supplementary tables and figures from e.g. “Fig. S1” or “Table S1” to “Supplementary Fig. 1” or “Supplemental Table 1”.
- Functions and variables have been made Roman if they are multi-letter and superscripts and subscripts have been made Roman if they are not scalar variables.
- Abstract shortened to <150 words (was originally 166).
- Data and code availability statements added**.
 - “Data availability:
For reasonable request or queries about data detailing the Crown-of-thorns Starfish Control Program on the Great Barrier Reef, please email JR which will be redirected to the data custodian.”
 - “Code availability:
For reasonable queries about the model code please email JR.”

** For review of the manuscript, code can be made available to the Editor and Reviewers upon request. However, separate permissions will need to be sought to run the model as data agreements do not allow sharing in this way. We may be able to request permissions to share the input files for that purpose only, but as we are not the data custodian, would need to separately seek permission. The code forms part of CSIRO IP in an ongoing project.

REVIEWERS' COMMENTS

Reviewer #4 (Remarks to the Author):

I just finished reading the revised version of the MS, and I am satisfied by their revisions. While they did not include a BACI analysis (which admittedly would have been difficult and time consuming to do), they've provided a simulation-based workaround which maps well against their original work, and also suggest how future studies might use a BACI to further understand the effects of CoTS control in the GBR.